# DC/L-SIGN recognition of spike glycoprotein promotes SARS-CoV-2 trans-infection and can be inhibited by a glycomimetic antagonist

Michel Thépaut[1], Joanna Luczkowiak[2], Corinne Vivès[1], Nuria Labiod[2], Isabelle Bally[1], Fátima Lasala[2], Yasmina Grimoire[1], Daphna Fenel[1], Sara Sattin[3], Nicole Thielens[1], Guy Schoehn[1], Anna Bernardi[3], Rafael Delgado[2]*, Franck Fieschi[1]*

1 Univ. Grenoble Alpes, CNRS, CEA, Institut de Biologie Structurale, Grenoble, France, 2 Instituto de Investigación Hospital Universitario 12 de Octubre (imas12), Universidad Complutense School of Medicine, Madrid, Spain, 3 Universita'degli Studi di Milano, Dipartimento di Chimica, Milano, Italy

☯ These authors contributed equally to this work.
* rafael.delgado@salud.madrid.org (RD); franck.fieschi@ibs.fr (FF)

## Abstract

The efficient spread of SARS-CoV-2 resulted in a unique pandemic in modern history. Despite early identification of ACE2 as the receptor for viral spike protein, much remains to be understood about the molecular events behind viral dissemination. We evaluated the contribution of C-type lectin receptors (CLR$_S$) of antigen-presenting cells, widely present in respiratory mucosa and lung tissue. DC-SIGN, L-SIGN, Langerin and MGL bind to diverse glycans of the spike using multiple interaction areas. Using pseudovirus and cells derived from monocytes or T-lymphocytes, we demonstrate that while virus capture by the CLRs examined does not allow direct cell infection, DC/L-SIGN, among these receptors, promote virus transfer to permissive ACE2$^+$ Vero E6 cells. A glycomimetic compound designed against DC-SIGN, enable inhibition of this process. These data have been then confirmed using authentic SARS-CoV-2 virus and human respiratory cell lines. Thus, we described a mechanism potentiating viral spreading of infection.

## Author summary

For their infectious effectiveness, viruses often use host attachment factors to improve their adhesion to the cell surface. This will mainly increase viruses concentration at cell surfaces, potentiating access and final engagement with their real entry receptors. This mechanism enhances viral infection of target cells or even allow viruses to be captured by non-permissive cells for secondary presentation to permissive cells by a process called trans-infection. While, the entry mechanism of SARS-CoV-2 using the ACE2 entry receptor, is well defined, little is known about additional factors explaining the high transmission rate of this virus. The level of glycosylations on the SARS-CoV-2 Spike protein prompted us to assess whether CLRs of immune cells, regularly diverted by pathogens, could play a role in the spread of SARS-CoV-2. Here, we show that these receptors are

**Data Availability Statement:** All numerical data are available from the OSFHOME database, at following url: https://osf.io/63jwr/.

**Funding:** This work used the platforms of the Grenoble Instruct-ERIC centre (ISBG; UMS 3518 CNRS-CEA-UGA-EMBL) within the Grenoble Partnership for Structural Biology (PSB), supported by FRISBI (ANR-10-INBS-05-02) and GRAL, within the University Grenoble Alpes graduate school CBH-EUR-GS (ANR-17-EURE-0003). The EM facility is supported by the Auvergne-Rhône-Alpes Region, the Fondation Recherche Medicale, the fonds FEDER and the GIS-Infrastructures Biologie Sante et Agronomie (IBISA). F.F. acknowledges the French Agence Nationale de la Recherche PIA for Glyco@Alps (ANR-15-IDEX-02). Research in R.D. lab is supported by grants from the Instituto de Investigación Carlos III, ISCIII, (FIS PI 1801007), the European Commission Horizon 2020 Framework Programme: Project VIRUSCAN FETPROACT-2016: 731868, and by Fundación Caixa-Health Research (Project StopEbola). The funders had no role in study design, data collection and analysis, decision to publish, or preparation of the manuscript.

**Competing interests:** The authors have declared that no competing interests exist.

able to recognize Spike envelope protein of SARS-CoV-2 and two receptors among the four tested, DC-SIGN and L-SIGN, are able to promote virus trans-infection.

This work identifies a new family of alternative SARS-CoV-2 cell receptors involved in uncharacterized dissemination mechanism. Moreover, considering the role of CLRs in immunomodulation, their early involvement opens avenues for understanding the imbalanced innate immune response observed in COVID-19 pathogenesis.

## Introduction

C-type Lectin Receptors (CLRs) are Pathogen Recognition Receptors (PRRs) involved in the detection of carbohydrate-based pathogen-associated molecular patterns by antigen-presenting cells (APCs), including macrophages and dendritic cells (DCs), and in the elaboration of the immune response [1,2]. Many innate immune cells express a wide variety of CLRs, which differ between cell types, allowing specific adjustments of the immune response upon target recognition. Thus, CLRs such as Dectin-2, Mincle, MGL (Macrophage galactose lectin), Langerin and DC-SIGN (Dendritic Cell-Specific Intercellular adhesion molecule-3-Grabbing Non-integrin) are major players in the recognition of pathogenic fungi, bacteria, parasites and viruses [3–6]. The interaction of these CLRs with their ligands allows DCs to modulate the immune response towards either activation or tolerance [7]. This is done through antigen presentation in lymphoid organs (primary mission of APCs) but also through the release of cytokines [8]. Thus, DCs have a major role in modulating the immune response from the early stages of infection. To fulfill their sentinel function, DCs are localized at and patrol the sites of first contact with a pathogen, such as epithelia and mucous interfaces, including the pulmonary and nasopharyngeal mucosae. Similarly, alveolar macrophages are found in the lung alveoli.

In this battle for infection, some pathogens have evolved strategies to circumvent the role of CLRs in activating immunity and even to divert CLRs to their benefit. Many enveloped viruses associate with CLRs and other host factors at the cell surface to facilitate the transfer towards their specific entry receptors that will trigger fusion of viral and host membranes. This kind of viral subversion has been reported for several CLRs, including L-SIGN (liver/lymph node-specific intracellular adhesion molecules-3 grabbing non-integrin, also called DC-SIGNR for DC-SIGN Related) and especially DC associated DC-SIGN, which promotes cis- and/or trans-infection of several viruses such as HIV, Cytomegalovirus, Dengue, Ebola and Zika [9–13]. In particular, DC-SIGN mediates direct HIV infection of DCs (cis-infection) and also induces trans-infection of T cells, the primary target of the virus [14], while in the case of Dengue and Ebola, DC-SIGN allows direct cis-infection of the receptor-carrying cells (Alvarez et al., 2002; Navarro-Sanchez et al., 2003). DC-SIGN and L-SIGN (herein after collectively referred to as DC/L-SIGN) have also been reported to be involved in the enhancement of SARS-CoV-1 infection [15–17].

In the context of the current COVID-19 pandemic, attention is now focused on the SARS-CoV-2 virus [18,19]. Coronaviruses use a homotrimeric glycosylated spike (S) protein protruding from their viral envelope to interact with cell membranes and promote fusion upon proteolytic activation. In the case of SARS-CoV-2, a first cleavage occurs at the level of a furin site (S1/S2 site), generating two functional subunits S1 and S2 that remain complexed in a pre-fusion conformation in newly formed virus. S2 contains the fusion machinery of the virus, while the surface unit S1 contains the receptor-binding domain (RBD) and stabilizes S2 in its pre-fusion conformation. The S protein of both SARS-CoV-2 [19–22] and SARS-CoV-1 [23]

uses ACE2 (Angiotensin-Converting Enzyme 2) as their entry receptor. For SARS-CoV-2 spike, interaction of its RBD with ACE2, as well as a second proteolytic cleavage at a S2' site, triggers further irreversible conformational changes in S2. This will engage the viral fusion process [20].

The sequence of events around the S protein/ACE2 interaction are becoming increasingly clearer, but much remains to be unraveled about additional factors facilitating the infection such as SARS-CoV-2 delivery to the ACE2 receptor. Indeed, S proteins from both SARS-CoV-1 and SARS-CoV-2 have identical affinity for ACE2 [22], but show very different transmission rates [24]. We posit that the enhanced transmission rate of SARS-CoV-2 relative to SARS--CoV-1 [25] might result from an efficient viral adhesion through host-cell attachment factors, which may promote efficient infection of ACE2$^+$ cells. This type of mechanism is frequently exploited by viruses to concentrate and scan cell surface for their receptor [26]. Additionally, in the case of SARS-CoV-2, a new paradigm is needed to untangle the complex clinical picture, resulting in a vast range of possible symptoms and in a spectrum of disease severity associated on one hand with active viral replication and cell infection through interaction with ACE2 along the respiratory tract, and, on the other hand, to the development of excessive immune activation, i.e. the so called "cytokine storm", that is related to additional tissue damage and potential fatal outcomes [27].

In this framework, C-type lectin PRRs and the APCs displaying them, i.e. DCs and macrophages, can play a role both as viral attachment factors and in immune activation. Thus, we focused on DC-SIGN and L-SIGN because of their involvement in SARS-CoV-1 infections [15–17]. L-SIGN is expressed in type II alveolar cells of human lungs as well as in endothelial cells and was identified as a cellular receptor for SARS-CoV-1 S glycoprotein [15]. DC-SIGN was also characterized as a SARS-CoV-1 S protein receptor [16] able to enhance cell entry by DC transfer to ACE2$^+$ pneumocytes [17].

Recent thorough glycan and structural analyses comparing both SARS-CoV-1/2 spike glycoproteins have shown that glycosylation is mostly conserved in the two proteins, both in position and nature of the glycans exposed [28–30], creating a glycan shield which complicates neutralization by antibodies. Secondly, elegant molecular dynamic simulations suggested how some of the spike glycans may directly modulate the dynamics of the interaction with ACE2, stabilizing the up conformation of the RBD domain [31,32]. Finally, and yet unexplored, spike glycans may contribute to infectivity by acting as anchor points for DC-SIGN and L-SIGN on host cells surfaces. Indeed, oligomannose-type glycans could therefore constitute ligands for CLRs and notably for DC-SIGN and L-SIGN. This argues also for the potential use of these CLRs by SARS-CoV-2, as does SARS-CoV-1. Additionally, some mutations modulating SARS-CoV-2 transmission have an impact on the glycosylation level of the spike. As an example, the D614G mutation has been reported as potentially increasing glycosylation at neighboring asparagine 616 [33–35]. A recent proteomic profiling study pointed to DC-SIGN as a mediator of genetic risk in COVID-19 [36] and finally DC/L-SIGN expression is induced by proinflammatory cytokines such as IL-4, IL-6, IL-10 and IL-13, known to be overexpressed in severe SARS and COVID-19 cases [37,38]. These observations prompted us to investigate the potential interaction of CLRs, notably DC/L-SIGN with SARS-CoV-2, through glycan recognition of the spike envelope glycoprotein, as well at their potential role in SARS-CoV-2 transmission.

## Results

### Production and stabilization of SARS-CoV-2 Spike Protein

In order to accurately analyze the interaction properties of the spike protein from SARS-CoV-2 with CLRs, we expressed and purified recombinant spike protein using an expression system

well-characterized in terms of its site-specific glycosylation. We used here the construct that was used 1) to obtain the cryo-electron microscopy structure [30] and 2) for extensive characterization of glycan distribution on the spike surface [28]. The spike protein was purified exploiting its 8xHis tag, followed by a Superose size exclusion chromatography (SEC). SEC chromatogram deconvolution allowed to select the best fractions (Fig 1B). SDS-PAGE analysis confirmed protein purity and differences in migration after reduction supported the presence of expected disulfide bridges and thus proper folding (Fig 1A). Furthermore, sample quality and trimeric assembly were confirmed by 2D class averages of the spike obtained from negatively stained sample observed under the electron microscope (Fig 1C). This construct contains "2P" stabilizing mutations at residues 986 and 987 [39], an inactivated furin cleavage site at the S1/S2 interface, and a C-terminal sequence optimizing trimerization [30]. Nonetheless, we observed a limited stability over a week time scale at 4˚C that was overcome by increasing the ionic strength of the storage buffer up to 500 mM NaCl, effectively preserving the trimeric state at 4˚C at least for three weeks (Fig 1D–1G). This "high-salt" concentration does not modify the structural properties of the protein as shown by identical elution profiles in SEC (Fig 1B); in addition, negative-stain EM images are better in "high-salt" conditions (Fig 1D and 1F).

## Several C-type lectin receptors interact with SARS-CoV-2 Spike Protein

DC-SIGN and L-SIGN have been described as receptor of SARS-CoV-1 and twenty out of the twenty-two SARS-CoV-2 S protein N-linked glycosylation sequons are conserved. Glycan shielding represent 60 to 90% of the spike surface considering the head or the stalk of the S ectodomain, respectively [31,40,41]. One third of N-glycans of SARS-CoV-2 spike are of the oligomannose type [28] and thus common ligands for DC-SIGN and L-SIGN, as well as for Langerin, a CLR of Langerhans cells, a subset of tissue-resident DCs of the skin, also present in mouth and vaginal mucosae [42].

SPR interaction experiments were performed with the various CLRs with immobilized SARS-CoV-2 S proteins. First, a S protein functionalized surface was generated using a standard procedure for covalent amine coupling onto the surface. The functionalization degree of this "non-oriented" surface depends upon the number of solvent exposed lysine residues (Fig 2A), which may be severely restricted by the glycan shield discussed above. Such restricted protein orientation could preclude the accessibility of some specific N-glycan clusters, located close to the linkage site and the sensor surface, thus hampering recognition by the oligomeric CLRs tested. To overcome these limitations, we devised and generated a so-called "oriented surface" where the S protein is captured *via* its C-terminal StrepTagII extremities onto a Streptactin functionalized surface (Fig 2B). In this set-up, no lateral parts of the S protein are attached to, and thus masked by, the sensor surface. Moreover, in the "oriented surface" set-up the spike protein is presented as it would be at the surface of the SARS-CoV-2 virus, which might better reflect the physiological interaction with host receptors. Hence, the "non-oriented" surface set up one may favor access to N-glycans of the spike's stalk domain while the "oriented" one may favor access to N-glycans of the head domain.

On the CLRs side, we tested exclusively recombinant constructs corresponding to the extracellular domains (ECD), containing both their carbohydrate recognition domain (CRD) and their oligomerization domain. Thus, the specific topological presentation of their CRD as well as their oligomeric status is preserved for each of the CLR, going from tetramers for DC-SIGN and L-SIGN to trimers for MGL and Langerin, ensuring interactions with avidity properties as close as possible to the physiological conditions for each CLR [43–46].

DC-SIGN and L-SIGN recognized the spike with the same profile on both surfaces (Fig 2A and 2B). Langerin, tested on the oriented surface, was found to interact with the S protein in

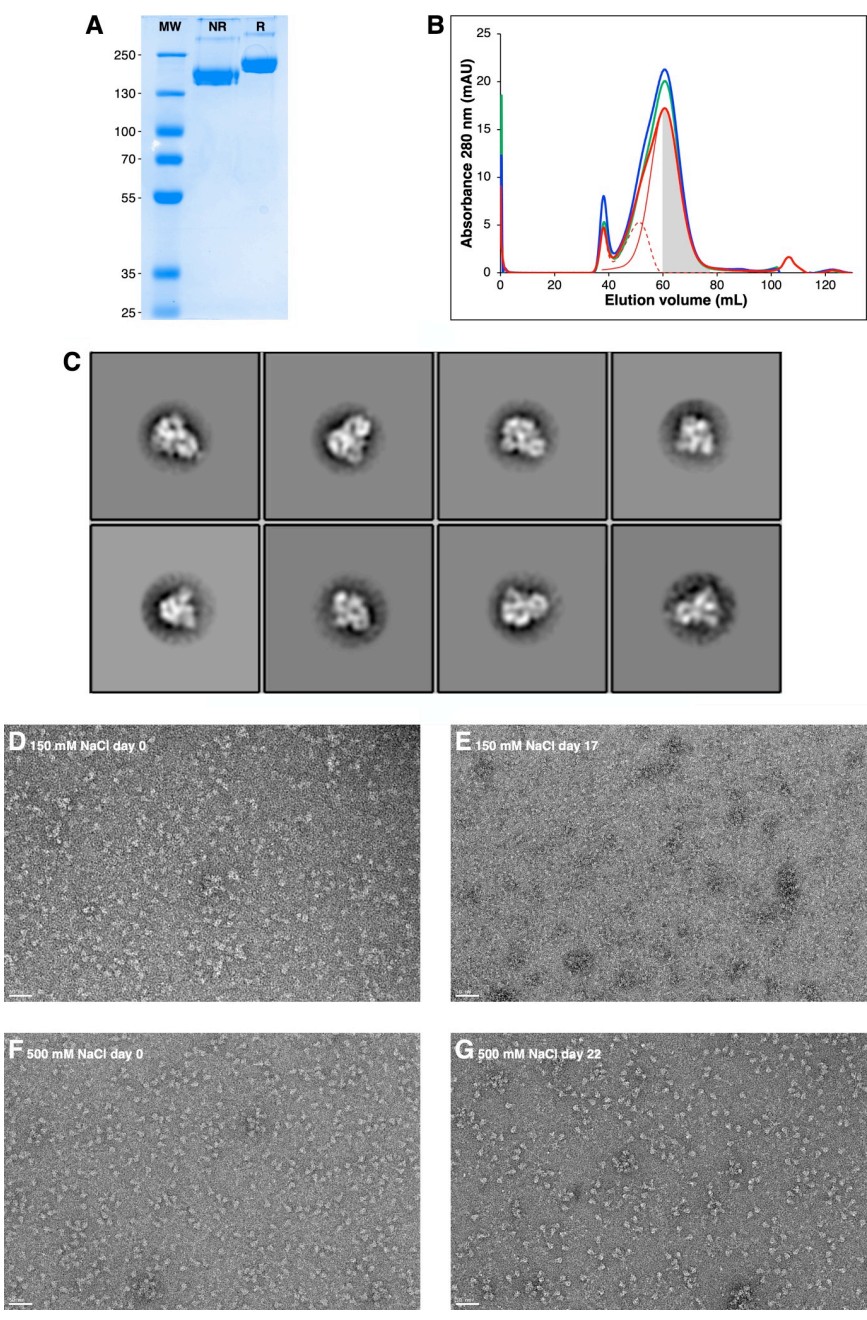

**Fig 1. Production and optimization of spike protein.** (**A**) SDS-PAGE analysis of 2 μg purified SARS-Cov-2 S protein; non-reduced and reduced, **NR** and **R**, respectively. (**B**) SEC Chromatograms of SARS-CoV-2 spike using buffer with 150 mM NaCl (green line), 375 mM NaCl (blue line) and 500 mM NaCl (thick red line). Manual deconvolution of gel filtration chromatogram at 500 mM NaCl: principal peak (thin red line) and contaminants (dashed red line). Collected fractions are represented by the grey area. (**C**) Classification of 2543 particles of SARS-Cov-2 spike after the purification step on HisTrap HP column, using Relion (auto-picking mode). (**D**) to (**G**) Quality control of SARS-CoV-2 S protein performed by negative staining Transmission Electron Microscopy (TEM) using uranyl acetate as stain (2% w/v). Scale bar is 50 nm. (**D**) and (**E**) Sample produced in 150 mM NaCl buffer, day of production and after 17 days at 4°C, respectively. (**F**) and (**G**) Sample produced in 500 mM NaCl buffer, day of production and after 22 days at 4°C, respectively.

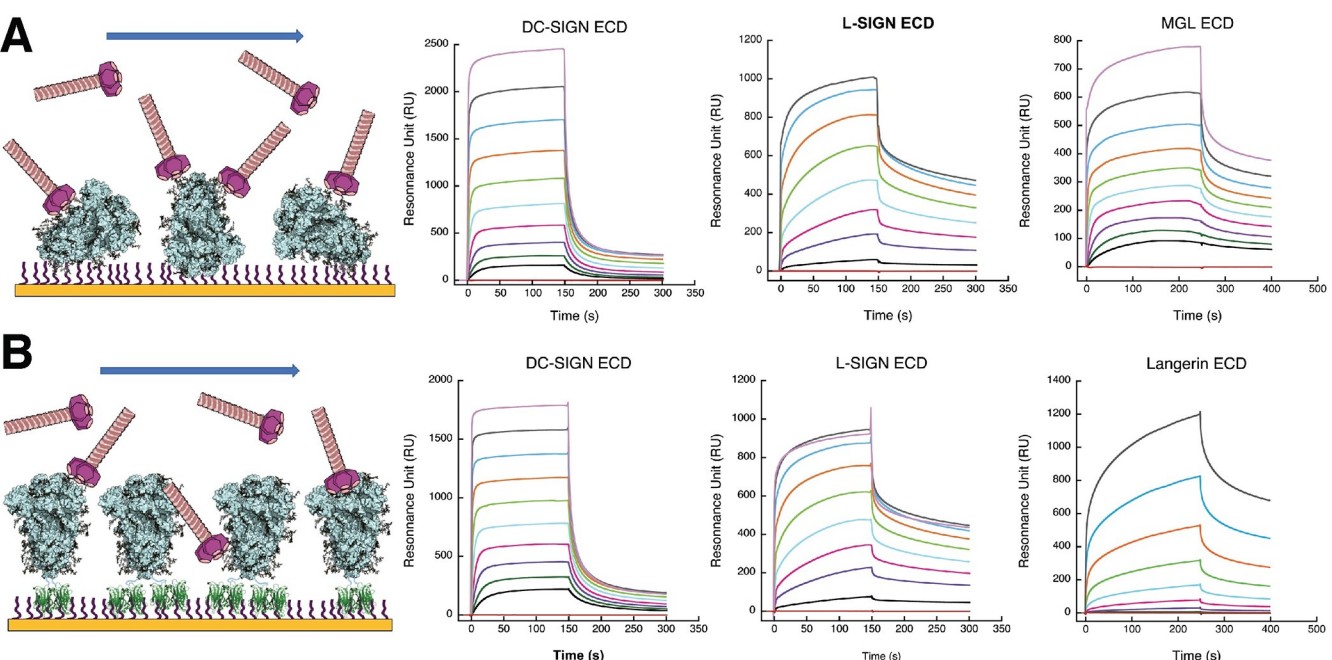

**Fig 2. CLRs interaction with SARS-CoV-2 S protein. (A)** Covalently functionalized S protein surface in random orientation. Multivalent CLRs oligomeric ectodomains (ECDs) are injected over the surface. From left to right, surfaces titrations were performed with ECDs of DC-SIGN, L-SIGN and MGL. **(B)** Oriented surface set-up. Surfaces were first functionalized with streptactin (in green) and then S protein was captured by its C-terminal double-StrepTagII. From left to right, surfaces titrations were performed with ECDs of DC-SIGN, L-SIGN and Langerin. The same color code for concentrations of all CLR-ECDs has been used for all panels. Concentrations injected (i.e. DC-SIGN ECD in A) range from 50 μM to 0.1 μM by 2-fold serial dilutions (decreasing concentrations from top to bottom). The red sensorgram, lower flat line, corresponds to control experiments with buffer injection with no CLR.

agreement with the presence of oligomannose-type glycans (Fig 2B) while MGL, a lectin that specifically recognizes glycans bearing terminal Gal or GalNAc residues, interacted with the S protein in the "non-oriented" set-up (Fig 2A). This shows that complex N-glycans may also serve as potential anchors for the SARS-CoV-2 S protein to cell surface CLRs.

While all CLRs tested interacted with the spike, the interactions observed are not all equivalent. Unfortunately, the complexity of the process involving probably multiple binding sites per oligomeric CLR prevented a kinetic fitting using classical kinetic models, which precluded the determination of kinetic rate constants. Nevertheless, an apparent equilibrium dissociation constant ($K_D$) could be obtained by steady state fitting for DC-SIGN, L-SIGN and MGL. For Langerin, despite a longer injection time, a much higher range of concentration would have been required to reach the equilibrium and accurately evaluate an apparent $K_D$. DC/L-SIGN and MGL showed affinities in the μM range, from around 2 to 10 μM (Table 1), depending on the CLR and the surface type, while Langerin has an affinity of at least one order of magnitude

**Table 1. Steady state determination of apparent $K_D$ for CLRs interaction with SARS-CoV-2 S protein.**

|  | DC-SIGN | L-SIGN | MGL | Langerin |
|---|---|---|---|---|
|  | μM | μM | μM |  |
| Non-oriented | 11.90 ± 4.6 | 1.80 ± 0.12 | 7 ± 1.5 | - |
| Oriented | 4.33 ± 0.7 | 1.66 ± 0.12 | - | n.d. |

Values reported results from steady state fitting of titration experiments as shown in Fig 2. They are the average from 2 to 4 independent measurements with different S protein preparations.

lower. Despite the impossibility to evaluate kinetic association and dissociation rate constants ($k_{on}$ and $k_{off}$), visual inspection of the sensorgrams clearly reveals different behaviors between DC-SIGN and L-SIGN independent of the surface set-up. While association and dissociation seem to be very fast for DC-SIGN, L-SIGN sensorgrams suggest a much slower association and dissociation rate that compensate each other to provide a $K_D$ similar to that of DC-SIGN. However, while the higher $k_{on}$ value for DC-SIGN argues for a faster formation of the DC-SIGN/S protein complex, the lower $k_{off}$ value for L-SIGN suggests that the L-SIGN/S-protein complex might be more stable.

Finally, for DC-SIGN and L-SIGN, which have been tested both on "non-oriented" and "oriented" S surface, no obvious differences have been observed in the interaction sensorgrams. This suggests that the interaction is not restricted to a limited glycan cluster, but rather that oligomannose-type glycans are multiple, accessible and distributed over the whole S protein.

## DC-SIGN forms multiple complexes with SARS-CoV-2 Spike Protein

The SPR interaction analysis argues for multiple potential binding sites for CLRs on the S protein. Such initial host adhesion mechanism could be essential for efficient viral capture, viral particles concentration on the cell surface and subsequent enhanced ACE2 targeting and infection. Negative stain electron microscopy was used to visualize potential DC-SIGN/S protein complexes.

Extemporaneously after a fresh purification of S protein, SEC fractions corresponding to the pure trimeric spike were mixed with a DC-SIGN ECD preparation in a molecular ratio 1:3 (1 trimeric spike for 3 tetrameric DC-SIGN ECD). In order to enrich the proportion of complex in the sample for EM observation, we reinjected this mix onto the same SEC column and recovered eluted fractions corresponding to higher molecular weight, and thus potentially to DC-SIGN/S protein complex (S1 Fig). These fractions were immediately used to prepare negatively stained electron microscopy grid (Fig 3). Fig 3A and 3B show control images of the DC-SIGN and spike sample used and images of the sample corresponding to the enriched complexes are shown in Fig 3C where DC-SIGN/S complexes can be clearly seen. Most images show a 1:1 complex but, in some cases, (Fig 3C, left) at least two molecules of DC-SIGN interacting with the spike protein can be detected. Even if it is not possible to precisely define the interacting region on the spike protein, several areas seem to be involved in the interaction with DC-SIGN.

## Antigen-Presenting Cells expressing DC-SIGN, MDDCs and M2-MDM are not infected by SARS-CoV-2 pseudovirions

To study the role of DC/L-SIGN in SARS-CoV-2 infection, VSV/SARS-CoV-2 pseudotyped viruses were firstly used for direct infection of primary monocyte-derived cells, including monocyte-derived DCs (MDDCs) and M2 monocyte-derived macrophages (M2-MDM), that have been shown to express DC-SIGN. As a control, VSV/EBOV-GP pseudotypes, expected to display enhanced infection in the presence of DC/L-SIGN receptor was used as well as VSV/VSV-G as a DC/L-SIGN independent pseudotype. To evaluate the infection mediated exclusively by DC/L-SIGN receptor on primary monocyte-derived cells we also examined the infection with these three pseudotypes in the presence of anti-DC/L-SIGN antibody.

VSV/SARS-CoV-2 did not infect MDDCs or M2-MDMs, despite DC-SIGN expression (Fig 4A). As expected, both primary cell lines were efficiently infected with VSV/EBOV-GP and this infection could be substantially blocked by an antibody targeting DC-SIGN. Infection of primary cells with EBOV does not exclusively depend on the presence of DC-SIGN on the cell

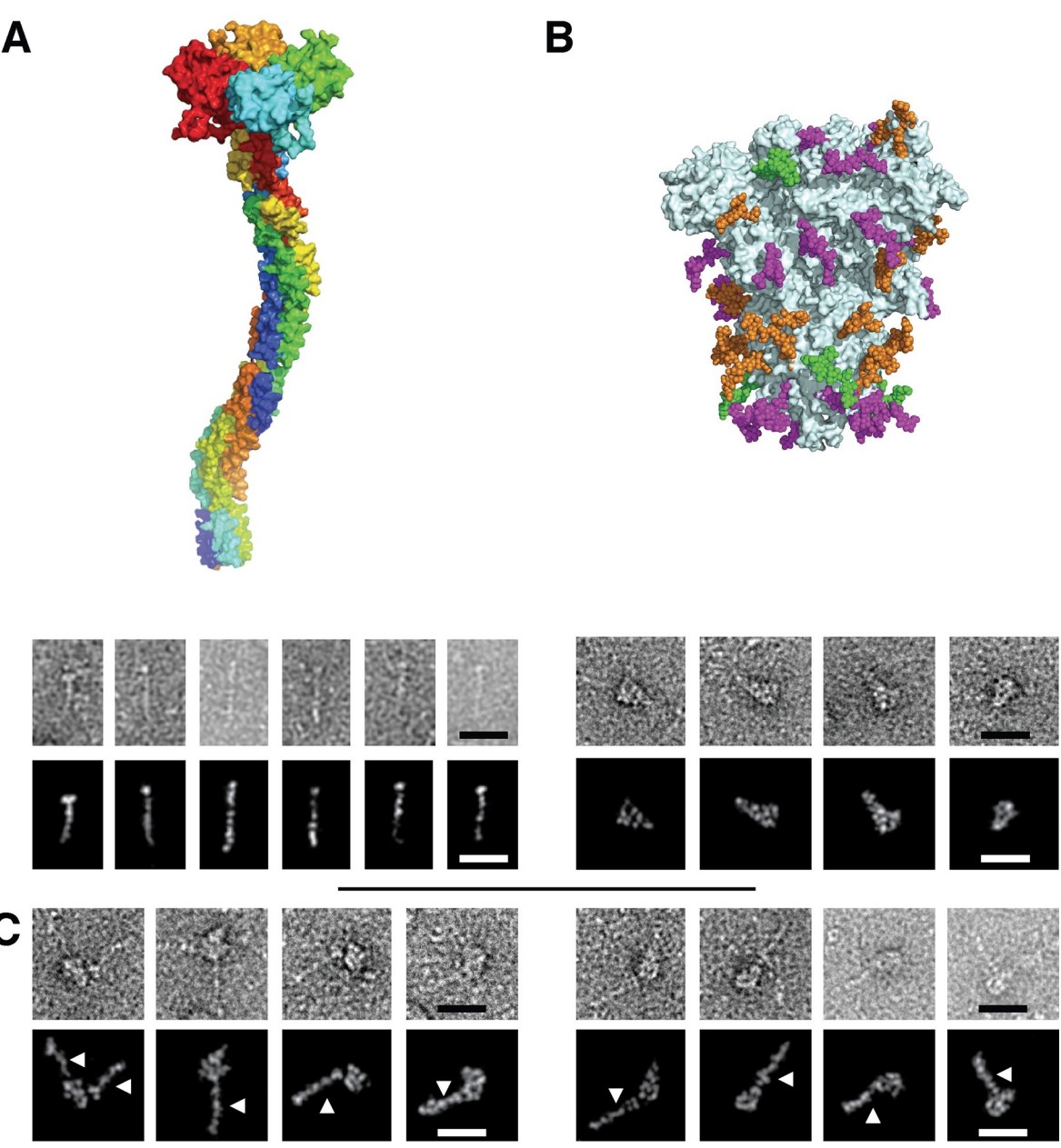

**Fig 3. Electron Microscopy micrographs of DC-SIGN/S protein complexes. (A)** DC-SIGN. Top: model of DC-SIGN ECD tetramer adapted from Tabarani et al (2009). Bottom: Negative staining images of DC-SIGN. Top row: original images; bottom row: Photoshop processed images. **(B)** Spike protein. Top: model of the glycosylated Spike adapted from model of Casalino et al and pdb 6vsb. Glycan sites are represented with color code derived from the work of Crispin *et coll.* [28], according to oligomannose-type glycan content, in green (80–100%), orange (30–79%) and magenta (0–29%). Bottom: Negative staining images of spike protein. Top row: original images, bottom row: Photoshop processed images. **(C)** Complex between DC-SIGN and spike protein. Negative staining image of the complexes between DC-SIGN and spike protein. The white arrows highlight DC-SIGN molecules. Top row: original images; bottom row: Photoshop processed images. In all cases the scale bar represents 25 nm.

surface, since other receptors are also responsible for the direct infection with EBOV [47] which explains the residual infection observed in the presence of an anti-DC/L-SIGN. DC-SIGN-mediated cis-infection was clearly blocked with anti-DC/L-SIGN in the case of MDDCs (92.5% inhibition of infection), followed by M2-MDM (68.4% inhibition). VSV/

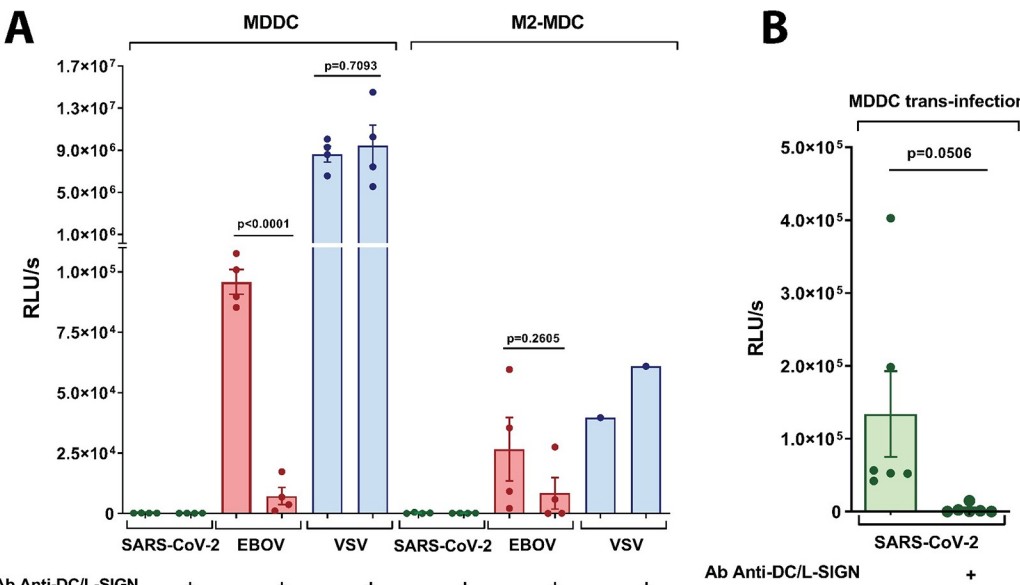

**Fig 4. SARS-CoV-2 cis- and trans-infection of monocytes-derived antigen presenting cells. (A)** Direct infection of primary cells: monocyte-derived dendritic cells (MDDCs), and M2 monocyte-derived macrophages (M2-MDM) with SARS-CoV-2, EBOV-GP and VSV-G pseudovirions. Bars represent mean ± SEM of mean of two independent experiments with cells from different donors performed in duplicates. Individual results are presented as scattered dot plot. As a control of inhibition of infection mediated by DC-SIGN, anti-DC-SIGN Ab was used. **(B)** MDDCs trans-infection of SARS-CoV-2 pseudovirions onto Vero E6 cells. Bars represent mean ± SEM of three independent experiments with cells from different donors performed in duplicates. Individual results are presented as scattered dot plot. As a control of inhibition of trans-infection mediated by DC-SIGN, anti-DC-SIGN Ab was used. Infection values are expressed as Relative Light Units (RLUs). Statistical significance was calculated by t-test.

VSV-G also showed a great infectivity of all primary cell lines. However, this infection was DC-SIGN independent, since anti-DC-SIGN antibodies did not impact the infection level (Fig 4A).

## MDDCs promote DC-SIGN-mediated trans-infection of SARS-CoV-2 pseudovirions

With other viruses, DC/L-SIGN are known to enhance viral uptake for direct infection in the process referred to as cis-infection or to allow subsequent transfer to susceptible cells in the process recognized as trans-infection [9,11]. To study the potential function of DC/L-SIGN in SARS-CoV-2 trans-infection, MDDCs were incubated with VSV/SARS-CoV-2 and placed onto susceptible Vero E6 cells, the reference ACE2$^+$ cell line for SARS-CoV-2 cell culture [19]. Interestingly, DC-SIGN promoted efficient SARS-CoV-2 trans-infection from MDDC to Vero E6 (Fig 4B). An anti-DC-SIGN antibody could reduce substantially the infectivity observed (98% inhibition), confirming the role of this CLR in the process of SARS-CoV-2 trans-infection.

## DC/L-SIGN but not Langerin mediate trans-infection of SARS-CoV-2 pseudovirions in a T-lymphocyte cell line

The role of DC/L-SIGN and Langerin in SARS-CoV-2 infection was also examined in Jurkat cells (a T-lymphocyte cell line lacking ACE2 expression). Parental Jurkat cell line, which does not express DC-SIGN, was used as control of infectivity along with Jurkats stably expressing DC/L-SIGN and Langerin [9]. VSV/VSV-G pseudovirions were used in parallel. VSV-G cell

entry is independent of the presence of CLRs and can efficiently infect Jurkat cells. Direct infection assay with VSV/SARS-CoV-2 showed no infectivity of any of the cell lines tested, Jurkat, Jurkat DC-SIGN and Jurkat L-SIGN (Fig 5A), indicating that DC/L-SIGN do not function as direct receptors for SARS-CoV-2. The use of DC/L-SIGN and Langerin by SARS-CoV-2 was further verified in trans-infection assays using Jurkat, Jurkat DC-SIGN, Jurkat L-SIGN and Jurkat Langerin cells. VSV/EBOV-GP trans-infection based on Jurkat cells is absolutely dependent on the presence of DC/L-SIGN or Langerin on the cell surface, since EBOV does not infect T-lymphocytes [48]. On the other hand, VSV/VSV-G is DC/L-SIGN or Langerin independent. Jurkat, Jurkat DC-SIGN, Jurkat L-SIGN and Jurkat Langerin cells were incubated with VSV-based pseudotypes and co-incubated upon susceptible Vero E6 cells monolayers. Similarly, to the results obtained with primary cells, we observed that DC/L-SIGN, but not Langerin, promoted efficient SARS-CoV-2 trans-infection of Vero E6 (Fig 5B). Anti-DC/L-SIGN antibodies could significantly reduce DC/L-SIGN-mediated trans-infection (86.7% and 78.7% inhibition, respectively) confirming the important role of these receptors in the SARS-CoV-2 trans-infection process.

Interestingly, no trans-infection was observed using Jurkat Langerin cells (Fig 5B). As expected, VSV/EBOV-GP achieved high DC/L-SIGN-mediated trans-infection which could be blocked by anti-DC/L-SIGN antibody (96.9% and 97.3% inhibition of infection respectively). VSV/EBOV-G uses Langerin as an attachment factor for trans-infection and this process could be partially blocked by anti-Langerin antibody (57.2% inhibition of infection). On the other hand, VSV/VSV-G does not use CLRs for trans-infection, thus no transmitted infection was detected in Vero E6 cells. The ratio of trans-infection mediated by DC/L-SIGN was calculated for VSV/SARS-CoV-2, VSV/EBOV-GP and VSV/VSV-G using the parental Jurkat cell line as reference. The ratio of VSV/SARS-CoV-2 trans-infection using Jurkat DC-SIGN and Jurkat L-SIGN was 408X and 680X respectively. VSV/EBOV-GP trans-infection ratio was 73X for Jurkat DC-SIGN and 549X for Jurkat L-SIGN (Fig 5C).

## DC-SIGN binding to the S protein and DC-SIGN-dependent trans-infection are inhibited by a known glycomimetic ligand of DC-SIGN (PM26)

Polyman26 (PM26, Fig 6A) is a multivalent glycomimetic mannoside tailored for optimal interaction with DC-SIGN [49]. It is known to bind DC-SIGN carbohydrate recognition domain (CRD), eliciting a Th-1 type response from human immature monocyte derived dendritic cells [50]. It also inhibits DC-SIGN mediated HIV infection of $CD8^{+}$ T lymphocytes with an $IC_{50}$ of 24 nM [49].

PM26 was used in SPR competition experiments to inhibit DC-SIGN binding to immobilized spike protein, both in the oriented and non-oriented set-ups (Fig 6B and 6C). The lectin (20 μM) was co-injected with variable concentrations of PM26 (from 50 μM to 0.1 μM), and the results showed clear dose-dependent inhibition. No significant difference was observed between the oriented and non-oriented surface, which is consistent with the binding data previously discussed (Fig 2). An $IC_{50}$ of 9.6±0.4 μM is estimated for the inhibition, which correlates with the apparent $K_{D}$ of DC-SIGN for the spike surfaces, which is here the reporting interaction (Fig 6C). In such competition test, it suggests that a higher avidity, towards DC-SIGN, can be awaited for PM26 [51].

The possibility of blocking DC-SIGN-mediated SARS-CoV-2 trans-infection using PM26 was thus examined. Jurkat DC-SIGN were pre-incubated with PM26 for 20 min before being challenged with VSV/SARS-CoV-2. The results of blocking DC-SIGN receptor by PM26 are shown as a percentage of inhibition of infection transmitted by Jurkat DC-SIGN to susceptible

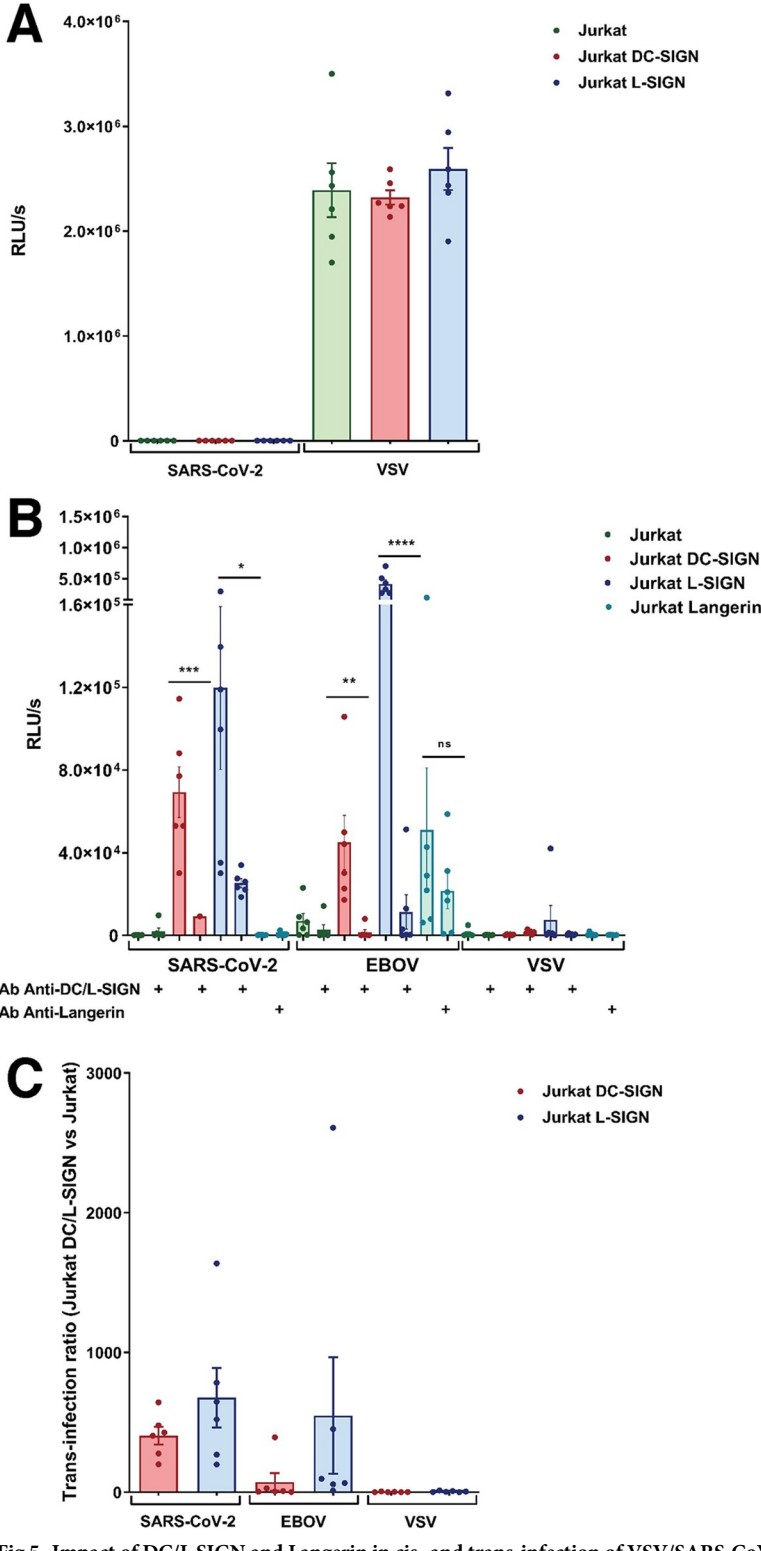

**Fig 5. Impact of DC/L SIGN and Langerin in cis- and trans-infection of VSV/SARS-CoV-2. (A)** Direct infection of Jurkat, Jurkat DC-SIGN and Jurkat L-SIGN with SARS-CoV-2 and VSV pseudovirus. **(B)** Trans-infection using Jurkat cell with or without CLRs DC-SIGN, L-SIGN and Langerin with SARS-CoV-2, EBOV and VSV pseudovirions on Vero E6 cells. As a control of inhibition of trans-infection mediated by CLRs, Ab specifically directed toward the corresponding CLR were used. **(C)** Trans-infection ratio of Jurkat DC-SIGN or Jurkat L-SIGN versus Jurkat cells with

SARS-CoV-2, EBOV and VSV pseudovirions on Vero E6 cells. Bars represent mean ± SEM of mean of two independent experiments performed in triplicates. Individual results are presented as scattered dot plot. Infection values from luciferase assays are expressed as Relative Light Units (RLUs) and ratios represents RLUs obtained by trans-infection with Jurkats expressing CLRs divided by those obtained from parental Jurkats. Statistical significance was calculated in panel B by t-test ($^*p<0.05$, $^{**}p<0.01$, $^{***}p<0.001$, $^{****}p<0.0001$).

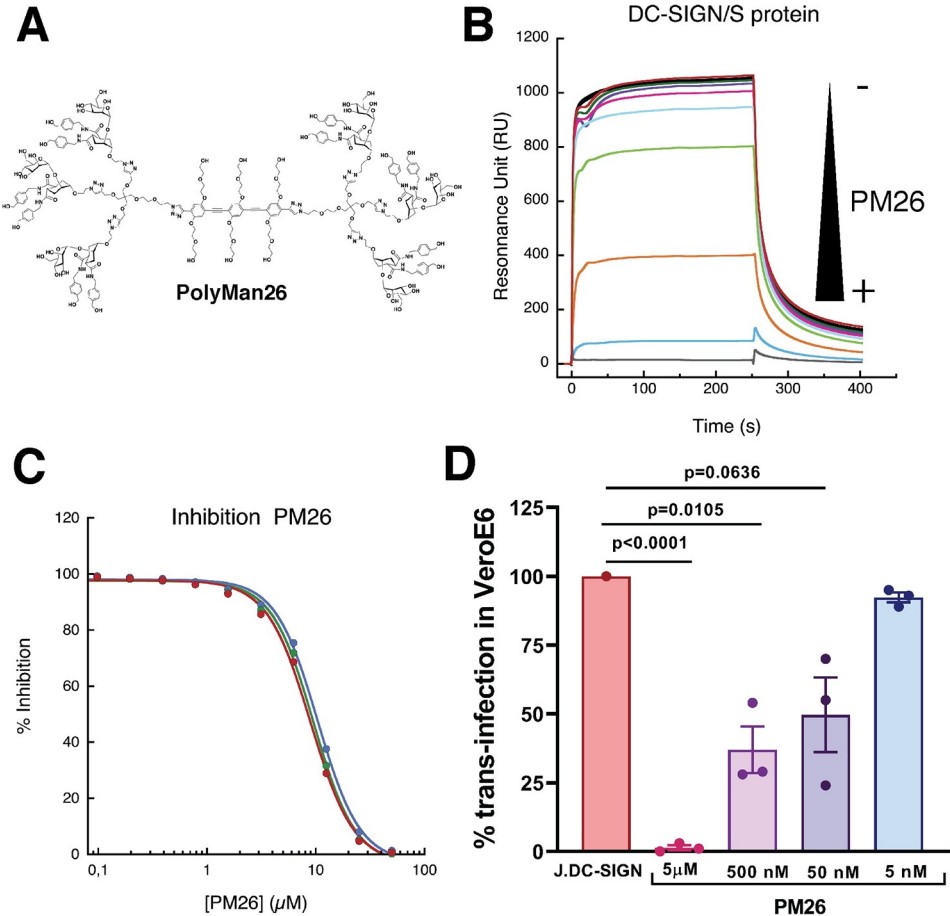

**Fig 6. PM26, a specific antagonist of DC-SIGN, can inhibit DC-SIGN/S protein interaction and trans-infection of VSV/SARS-CoV-2. (A)** Structure of Polyman26 (PM26). **(B)** Sensorgrams of DC-SIGN/S protein interaction inhibition by a range of PM26 concentration. DC-SIGN is injected at the constant concentration of 20 μM, PM26 is co-injected at decreasing concentration going from 50 μM to 0.1 μM. S protein is presented in an oriented mode in the experiment presented here. **(C)** PM26 inhibition curve of the DC-SIGN/spike interaction using two independent oriented (blue, and green) or non-oriented spike surface (red) set up as described in Fig 2. The blue curve of inhibition derives from the sensorgrams presented in (B). Sensorgrams corresponding to the two others inhibitions curves are presented in the S2 Fig. **(D)** Inhibition of trans-infection of Jurkat DC-SIGN with SARS-CoV-2-pseudotyped rVSV-luc. Bars correspond to mean ± SEM of triplicates from an experiment of inhibition of SARS-CoV-2 trans-infection. Results are presented as percentage of SARS-CoV-2 trans-infection in the presence of PM26 as compared to trans-infection of SARS-CoV-2 in Vero E6 mediated by Jurkat DC-SIGN without addition of any compound. Trans-infection bar for Jurkat DC-SIGN established as 100% of the trans-infection correspond to the mean of triplicates of SARS-CoV-2 trans-infection by Jurkat DC-SIGN without addition of any compound.SARS-CoV-2-pseudotyped rVSV-luc was used at MOI: 1. Individual results are presented as scattered dot plot. Statistical significance was calculated in panel D by t-test ($^*p<0.05$, $^{**}p<0.01$, $^{***}p<0.001$, $^{****}p<0.0001$). The IC$_{50}$ value for PM26 was estimated using GraphPad Prism v8 with a 95% confidence interval and settings for normalize dose-response curves (IC$_{50}$ = 94 nM, 95%CI = 35 nM– 249 nM).

Vero E6, as compared with SARS-CoV-2 trans-infection mediated by Jurkat DC-SIGN to Vero E6 without addition of any compound. PM26 was tested at four different concentrations: 5 μM, 500 nM, 50 nM and 5 nM and demonstrated the $IC_{50}$ of 94 nM (95%CI = 35 nM– 249 nM) for the inhibition of VSV/SARS-CoV-2 trans-infection, which is consistent with the results described for HIV inhibition and confirming an effective affinity in the nanomolar range for PM26 [49].

## MDDCs promote DC-SIGN-mediated trans-infection of SARS-CoV-2 of the human lung cell line Calu-3

To confirm the role of DC/L-SIGN in SARS-CoV-2 trans-infection observed when using pseudotypes assay, we have also performed the trans-infection experiment using the authentic SARS-CoV-2 isolate. We challenged MDDCs or Jurkat DC-SIGN with SARS-CoV-2 for 2h and after exhaustive washing, we co-incubated MDDCs with susceptible Calu-3 cells. We used SARS-CoV-2 clinical isolate at MOI: 0.1–1 in case of MDDC and at MOI: 1 in case of Jurkat DC-SIGN. The trans-infection in Calu-3 was estimated by using an anti-SARS-CoV-2 Nucleocapsid antibody and colorimetric detection as described. As a positive control of infection, we performed direct infection of Calu-3 with equal titer of the SARS-CoV-2. To estimate the optical density values of blank for this experiment, we have co-incubated MDDCs or Jurkat DC-SIGN with Calu-3 cells in the same conditions however without challenging them with SARS-CoV-2. As a control of DC/L-SIGN contribution in trans-infection experiment we have used anti-DC/L-SIGN antibody. Additionally, we have performed the trans-infection assay in the presence of PBMC or parental Jurkat not expressing DC-SIGN.

Trans-infection experiment with the authentic SARS-CoV-2 showed that DC/L-SIGN promoted efficient trans-infection of Calu-3 mediated by both, MDDC and Jurkat DC-SIGN (Fig 7A and 7B). Antibody anti-DC/L-SIGN could reduce the DC-SIGN-mediated trans-infection

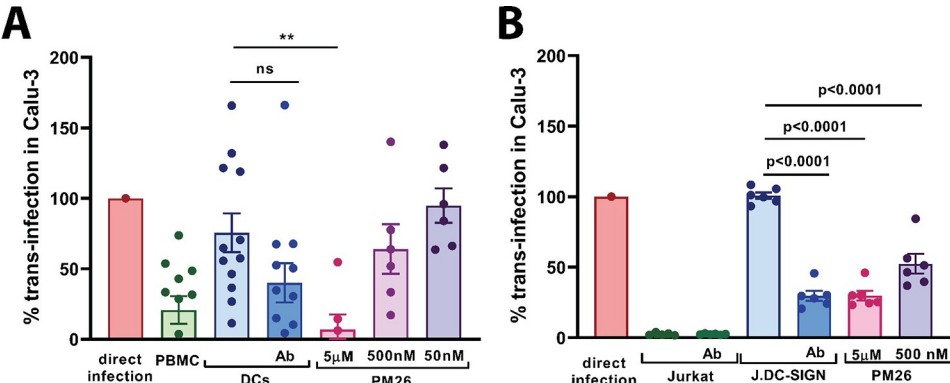

**Fig 7. DC-SIGN-mediated trans-infection of a respiratory cell line Calu-3 and T lymphocyte cell line, with the SARS-CoV-2 clinical isolate.** A. MDDCs or PBMCs trans-infection of Calu-3 with SARS-CoV-2. SARS-CoV-2 clinical isolate was used at MOI: 0.1–1. Compound PM26 was used at 5 μM, 500 nM and 50 nM. Direct infection bar established as 100% of the infection correspond to the mean of six replicates of direct infection of Calu-3 cells with the same MOI of the authentic SARS-CoV-2. MDDCs and PBMCs for this experiment were obtained from 4 different healthy blood donors. Bars represent mean of triplicates of two independent experiments (3 donors and one donor respectively) ± SEM of mean. As a control of inhibition of infection mediated by DC-SIGN, Ab anti-DC-SIGN was used. B. Jurkat and Jurkat DC-SIGN-mediated trans-infection of Calu-3 with the authentic SARS-CoV-2. SARS-CoV-2 was used at MOI: 1. Compound PM26 was used at 5 μM and 500 nM. Direct infection bar established as 100% of the infection correspond to the mean of triplicated of direct infection of Calu-3 cells with the same MOI of the authentic SARS-CoV-2. t test was performed to estimate the statistical significance of inhibition of infection as compared to DC-SIGN-expressing cells by either Ab anti-DC-SIGN or PM26. Individual results are presented as scattered dot plot. Statistical significance was calculated by t-test (*p<0.05, **p<0.01, ***p<0.001, ****p<0.0001).

mediated by MDDC and Jurkat DC-SIGN (p = 0.0832 and p<0.0001, respectively) confirming the essential function of this receptor in SARS-CoV-2 trans-infection process.

The potency of inhibiting the DC-SIGN-mediated trans-infection with the authentic SARS-CoV-2 isolate by the glycomimetic compound PM26 was studied by six replicates using both systems, MDDC and Jurkat DC-SIGN (Fig 7A and 7B). In MDDC-mediated trans-infection PM26 was tested at 3 concentrations: 5 µM (81% inhibition of the trans-infection, p = 0.0049), 500 nM (15.5% inhibition of the trans-infection) and 50 nM. In Jurkat DC-SIGN-mediated trans-infection PM26 was examined at 2 concentrations: 5 µM (70.4% inhibition of the trans-infection, p<0.0001) and 500 nM (60.6% inhibition of the trans-infection, p<0.0001).

## Discussion

Even if viruses target mainly one specific entry receptor within their infection cycle, their efficiency often largely depends upon additional binding events at the cell surface, which promote access to this receptor [4,52]. Although such additional receptors may not promote any fusion step, they can drive viral internalization through endocytic processes or simply by viral adhesion to the host cell, accumulation of viral particles on the cell surface and finally engagement with the entry receptor followed by the fusion event [53–55]. Different types of attachment factors can be found on the cell surface: either glycans, such as heparan-sulfate [56], glycolipids or protein N-glycans, often targeted by envelope viral protein with lectin-like properties [57], or immune lectin-type receptors including CLRs and Siglecs [58,59]. CLRs primary role is to sense the presence of pathogen through recognition of specific non self-glycans. However, some viruses are able to hijack them as co-receptors for cell entry or hiding. HIV represents a famous example since it exploits DC-SIGN in genital mucosa to promote uptake by DCs and T lymphocytes trans-infection [11].

SARS-CoV-1 and MERS-Co-V use heparan sulfate and sialic acid exposed at the cell surface to attach to cell membranes [60–62]. Remarkably, DC/L-SIGN were not only reported to act as attachment factors for SARS-CoV-1, but also as promoters of viral dissemination through trans-infection. Here we investigate whether those CLRs could also play a role in SARS-CoV-2 infection and dissemination [17,63]. While we were finalizing this work, a first preprint article described the binding (Elisa test) of Fc-CLR constructs, notably based on DC-SIGN and MGL, to a commercial RBD domain of the SARS-CoV-2 [64]. Given the importance of the role played by glycan determinants in this recognition event, peculiar attention must be paid to the quality of the S protein sample used. Indeed, it has to be ideally as close as possible to the physiological product in terms of glycosylation pattern and distribution. In particular, the expression system considered as well as the local protein environment may have a strong impact on the type of glycan added and their level of maturation [65]. Viral envelope proteins display a dense array of glycans resulting from evolutionary pressure to mask immunogenic epitopes at their surfaces. This glycan density coupled to specific structural features of envelope proteins generate steric constraints preventing proper access of glycan processing enzymes to some substrate glycans [65]. Expressing the whole spike ectodomain or just the single RBD domain may therefore lead to very different N-glycan distribution, especially considering that the RBD contains only 2 N-glycosylations sites, while up to 66 N-glycan sites are found over the whole spike protein [28]. For these reasons, we selected the entire ectodomain of S as our model to investigate additional attachment factors for SARS-CoV-2. We expressed the protein using the same construct enabling the spike EM structure [30] and its glycan profiling [28], using a HEK293-derived expression system known to provide glycosylation pattern similar to epithelial tissues. Similarly, we expressed the entire ectodomain for the CLRs as well, avoiding Fc-

CRD constructs, in order to preserve their specific oligomeric assembly and therefore their avidity properties.

Using SPR we showed that all the CLRs tested interact with the spike protein. Three of those, DC-SIGN, L-SIGN and Langerin, share the ability to recognize high-mannose oligosaccharides. In particular, L-SIGN is tightly specific for high-mannose, while DC-SIGN additionally recognizes fucose based ligands (several Lewis type glycans) and Langerin binds sulfated sugars as well. MGL is specific for Gal and GalNAc terminated glycans and may bind to complex N-glycans as a function of their level of maturation [66]. Analyzing the glycosylation pattern of the spike protein, reported in Fig 3B, all glycosylation sites depicted in green or orange are potential ligands for L-SIGN and Langerin, with different level of probability from site to site, while MGL's ligands will be found in magenta sites. DC-SIGN might potentially recognize all of them. Beside all considerations about specificity, the accessibility of N-glycan sites upon spike presentation at the SARS-CoV-2 virus surface is also of paramount importance for recognition. DC-SIGN and L-SIGN share the same tetrameric organization and they recognize with similar avidity the spike functionalized SPR surface, suggesting that they share a primary recognition epitope—i.e. high mannose. The SPR experiments described here were performed sequentially on the same spike surfaces with the different CLRs. Of these, DC-SIGN and L-SIGN have similar organization and molecular weight [67], thus the difference in RU level reached by the two lectins at their equilibrium (approx. 1000 RU higher for DC-SIGN) suggests that there is more DC-SIGN binding and thus more epitopes available for it, implying that high mannose are not the unique glycan epitope used here by DC-SIGN. Thus DC-SIGN can also bind to some other complex N-glycosylation sites (in magenta in Fig 3B), possibly presenting a proper fucosylation pattern that generates Lewis-type epitopes. These considerations, in addition to the oligomeric state of the CLRs examined, lead us to rule out a simple interaction with a single preferential epitope and a 1:1 stoichiometry in favor of a more complex picture with multiple and simultaneous binding events, like the "Velcro effect" often recalled when discussing glycan-protein interactions [51]. This is clearly supported by the EM characterization of Spike/DC-SIGN complexes (Fig 3C) that shows several interactions areas on the spike and can also explain the absence of affinity differences between non-oriented and oriented spikes surfaces in SPR.

The complexity of the binding event(s) described above does not allow to extract kinetic association and dissociation rates from the sensorgrams. Only a global apparent $K_D$ could be inferred, giving avidity levels. However, L-SIGN may have a slightly better affinity (around 2 μM, while values ranging from 5 to 10 μM have been obtained for DC-SIGN and MGL) and seems to generate more stable complexes. Such μM range of affinity, as determined here for soluble forms of CLR, will result in surface avidity of several orders of magnitude higher at the cell membrane [51]. Indeed, CLRs, as DC-SIGN, are organized in microdomains, allowing multiple attachment points for viral capture [68].

CLRs and particularly DC/L-SIGN have been associated to important steps of viral entry and infection of different viruses. Participation of DC-SIGN in the infectivity and initial dissemination of a number of viral agents has been described in animal models for measles [69,70], Japanese encephalitis virus [71] and in vivo for HIV-1 [11]. The founder viruses that initiate HIV infection through mucosa exhibit higher content of high-mannose carbohydrates [72], as well as higher binding to DCs dependent on DC-SIGN expression [73]. In the case of Ebola virus, DCs and macrophages have been shown to be the initial targets of infection in macaques [74,75] and circulating DC-SIGN+ DCs have been shown to be the first cell subset infected upon intranasal EBOV inoculation in a murine model [76].

In SARS-CoV-1 infection, DC/L-SIGN can enhance viral infection and dissemination [16,17] and even it has been proposed that L-SIGN could act as an alternative cell receptor to

ACE2 [15]. Our work shows that DC/L-SIGN are important factors contributing to additional routes of infection also mediated by the S protein of SARS-CoV-2. This trans-infection process greatly facilitates viral transmission to susceptible cells. *In vivo*, DC-SIGN is largely expressed in immature dendritic cells in submucosa and tissue resident macrophages, including alveolar macrophages [77] whereas L-SIGN is highly expressed in human type II alveolar cells and lung endothelial cells [15]. Using primary MDDCs and M2-MDM, two of the established primary cell models to explore DC-SIGN interactions [78], and a well-established VSV-based pseudo-virion system, we did not observe direct infection of MDDCs or M2-MDM, indicating that DC-SIGN expressed by these cells does not function as an alternative receptor (Fig 4A). MDDCs and M2-MDM, as expected, were infected by VSV pseudotyped with EBOV-GP or VSV-G, although in a DC-SIGN dependent and independent manner respectively. DC-SIGN on MDDC, however, showed a clear function as an enhancer of infection, in a process known as trans-infection [11] (Fig 4B). Similar results were obtained with the T-lymphocyte Jurkat cell line. T lymphocytes lack ACE2 expression [79] and both the parental Jurkat cell line and Jurkats expressing DC/L-SIGN were not directly infected by SARS-CoV-2 pseudovirions (Fig 5A). Therefore, we did not observe that these CLRs can function as alternative receptors to ACE2 in non-susceptible cells, such as T lymphocytes, as it has been recently suggested by Amraie et al. [80]. On the other hand, DC/L-SIGN expression on Jurkat cells allows binding of SARS-CoV-2 pseudovirions. Future studies will address the trans-infection mechanism and whether the virus is simply attached and stabilized at cell surfaces by CLRs or internalized in dedicated compartment awaiting trans-infective release. This trans-infection mechanism is significantly inhibited by a specific DC/L-SIGN antibody and, remarkably, it seems exclusive of DC/L-SIGN, since the related CLR Langerin does not mediate trans-infection in Jurkat cells. This is similar to what has been reported for HIV-1, since Langerin acts as an antiviral receptor that degrades HIV-1 via internalization and subsequent degradation [81]. On the other hand, Langerin appears to function as a trans-receptor for EBOV pseudovirions highlighting the complexity of CLRs recognition patterns and functionalities.

Although the pseudovirus system has been extensively validated for SARS-CoV-2 viral entry in neutralizing and pathogenic studies [82] we have confirmed our results with authentic SARS-CoV-2 using both MDDC and Jurkat-DC-SIGN cells (Fig 7A and 7B). Furthermore, the respiratory Calu-3 cell line was used to confirm SARS-CoV-2 DC-SIGN dependent trans-infection in a more physiological way showing both the relevance of the mechanism described and the effect of inhibition of the lectin binding properties with the glycomimetic DC-SIGN ligand PM26.

The biological relevance of the interaction of CLRs with SARS-CoV-2 remains yet to be fully explored. The innate immune system provides the first line of defense against viruses and it is now clear that severe COVID-19 is largely due to an imbalance between viral replication and antiviral and pro-inflammatory responses [83]. Here we demonstrate that DC/L-SIGN can function, in primary cells and cell lines, as potent trans-receptors. The expression of DC/L-SIGN in relevant cell subsets along the respiratory tract, such as submucosal DCs and Macrophages or importantly type II alveolar cells, which also express high level of ACE2, together with their potency to enhance viral infection could be critical for the pathogenesis of COVID-19. In this context, it is important to note that DC-SIGN expression is negatively regulated by IFN, TGF-β, and anti-inflammatory agents [84] and that DC/L-SIGN activation through ligand recognition triggers the production of pro-inflammatory cytokines such as IL-6 and IL-12 [58]. These cytokines are raised up in severe forms of COVID-19 and are among the mediators responsible for the development of the cytokine storm related to poor prognosis and eventual fatalities [85]. DC/L-SIGN expression can be upregulated as well, since it has been demonstrated that while innate immune responses are potently activated by SARS-CoV-2, it also counteracts the production of type I and type III interferon [86].

Innate immunity plays an important role in initial control and the pathogenesis of COVID-19. There are a number of clinical trials focused on the use of interferon and anti-inflammatory agents to try to modulate the inflammatory response triggered by SARS-CoV-2 [87]. Such approaches could be the key for treating severe cases of COVID-19, which correlate with cytokine storm and hyper-activation of immune responses [37]. On the other hand, indiscriminate blocking of immune signaling would be counterproductive in the early stages of the infection and for patients with moderate disease, who appear to maintain a functional, well-adapted immune response.

DC/L-SIGN antagonists may help reducing the severity of SARS-CoV-2 infection by inhibiting the trans-receptor role played by these CLRs. We show here that PM26, a glycomimetic antagonist of DC-SIGN, inhibits the interaction of the S protein with the lectin receptor and blocks DC-SIGN-mediated SARS-CoV-2 trans-infection of susceptible Vero E6 cells. PM26 is known to act both by binding DC-SIGN CRD and by promoting its internalization [50], thus reducing the lectin concentration on cell surface and further impairing the ability of the virus to exploit it for enhancing its dissemination. Additionally, upon binding to DC-SIGN PM26 was shown to induce a pro-inflammatory anti-viral response [50], which should be beneficial at the onset of the infection and may help to steer the immune response towards a profile correlated with milder forms of the disease.

Demonstration of a model of involvement of APCs in SARS-CoV-2 early dissemination through CRLs DC/L-SIGN, could open new avenues for understanding and treating the imbalanced innate immune response observed in COVID-19 pathogenesis. The potential of this and other CLRs antagonists in prevention or as part of combination therapy of COVID-19 needs to be further explored.

# Materials and methods

## Protein production and purification

The extracellular domains (ECD) of DC-SIGN (residues: 66–404), L-SIGN (residues: 78–399), Langerin (residues: 68–328) and MGL (residues: 61–292) were produced and purified as already described [88–91] while SARS-CoV-2 Spike protein was expressed and purified as follows. The mammalian expression vectors used for the S ectodomain, derived from a pαH vector, was a kind gift from J. McLellan [30]. This construct possesses, in its C-terminus, an 8xHis tag followed by 2 StrepTagII. EXPI293 cells grown in EXPI293 expression medium were transiently transfected with the S ectodomain vector according to the manufacturer's protocol (Thermo Fisher Scientific). Cultures were harvested five days after transfection and the medium was separated from the cells by centrifugation. Supernatant was passed through a 0.45 μm filter and used for a two-step protein purification on Aktä Xpress, with a HisTrap HP column (GE Healthcare) and a Superose 6 column (GE Healthcare). Before sample loading, columns were equilibrated into 20 mM Tris pH 7.4; supplemented with variable concentrations of NaCl (150–500 mM) depending on the experiments. Unbound proteins were eluted from affinity column with equilibration buffer, contaminants with the same buffer supplemented with 75 mM imidazole while the spike protein was eluted with equilibration buffer supplemented with 500 mM imidazole and immediately loaded onto a gel filtration column run in equilibration buffer. Fractions of interest were pooled and concentrated at 0.5 mg/mL on an Amicon Ultra 50K centrifugal filter according to the manufacturer's protocol (Millipore). Fractions and samples were analyzed for purity using SPS-PAGE at 8% acrylamide gel, with or without ß-mercapto-ethanol as reductant. The concentration of purified spike protein was estimated using an absorption coefficient ($A_{1\%,1cm}$) at 280 nm of 10.4 calculated using the PROTPARAM program (http://web.expasy.org/protparam/) on the Expasy Server. Quality

control and visualization of the different samples was performed by negative staining Transmission Electron Microscopy (TEM) using Uranyl Acetate as stain (2% w/v).

## Negative staining electron microscopy

Negative-stain grids were prepared using the mica-carbon flotation technique [92]. 4 µL of spike samples from purifications diluted at about 0.05–0.1 mg/mL were adsorbed on the clean side of a carbon film previously evaporated on mica and then stained using 2% w/v Uranyl Acetate for 30 s. The sample/carbon ensemble was then fished using an EM grid and air-dried. Images were acquired under low dose conditions (<30 e−/Å2) on a Tecnai 12 FEI electron microscope operated at 120 kV using a Gatan ORIUS SC1000 camera (Gatan, Inc., Pleasanton, CA) at 30,000x nominal magnification. To facilitate the visualization of the molecules, a Gaussian filter was applied to the images using Photoshop, then the gray levels were saturated and the background eliminated. For the 2D classification, images were processed with RELION 2.1 [93]. CTF was estimated with CTFFind-4.1 [94]. An initial set of 409 particles (box size of 512 pixels, sampling of 2.2 Å/pixel) was obtained by manual picking. After 2D classification the best looking 2D class averages were used as references for Autopicking. A set of around 35 000 particles (box size of 256 pixels, sampling of 4.4 Å/pixel, mask diameter 300Å) was obtained by Autopicking with a gaussian blob. The 8 best obtained classes were calculated from 2854 particles.

## SPR binding studies

Two types of surface plasmon resonance (SPR) experiments were performed at 25˚C on a Biacore T200. The first experiments with non-oriented spike surfaces were performed using a CM5 sensor chip, functionalized at 5 µL/min. Spike protein was immobilized on flow cells using a classical amine-coupling method. Flow cell 1 (Fc1) was functionalized with BSA and used as reference surface. Fc1 to 4 were activated with 50 µL of a 0.2 M EDC/ 0.05 M NHS mixture and functionalized with 20 µg/mL BSA (Fc1) or 50 µg/mL spike protein (Fc2-4), the remaining activated groups of all cells were blocked with 30 µL of 1 M ethanolamine pH 8.5. The four Fc were treated at 100 µL/min with 5 µL of 10 mM HCl to remove non-specifically bound protein and 5 µL of 50 mM NaOH/ 1M NaCl to expose surface to regeneration protocol. Finally, an average of 2500, 3000 and 2300 RU of spike protein were functionalized onto Fc2, 3 and 4, respectively.

The second type of experiments used oriented spike surfaces and were performed using a CM3 sensor chip functionalized at 5 µL/min. The procedure for oriented functionalization has been described in our recent work [51]. Fc4 was functionalized with non-oriented spike protein exactly as described for the CM5 sensor chip using 20 µg/mL spike protein (final functionalization of 1430 RU). Fc1 (reference surface) and Fc2 were activated with 50 µL of a 0.2M EDC/ 0.05 M NHS mixture and functionalized with 170 µg/mL StrepTactin (IBA company) and the remaining activated groups were blocked with 80 µL of 1 M ethanolamine. Flow cells were treated at 100 µL/min with 5 µL of 10 mM HCl and 5 µL of 50 mM NaOH/ 1M NaCl. An average of 2000 RU of covalently immobilized StrepTactin was obtained. The spike protein was then captured at 20 µg/mL on Fc2 via its StrepTags. The surface was washed at 100 µL/ min with 1M NaCl. Fc2 final level of functionalization was 990 RU. The presence of a double StrepTagII at the C-terminal extremity of the S protein (a total of 6 within the spike trimer) led to very stable surfaces and capture did not require the additional EDC/NHS treatment previously reported for such oriented functionalization. On both type of surfaces, for direct interaction studies, increasing concentrations of extracellular domain of DC-SIGN, L-SIGN, MGL and Langerin were prepared in a running buffer composed of 25 mM Tris pH 8, 150 mM

NaCl, 4 mM $CaCl_2$, 0.05% P20 surfactant, and either 80 μL of each DC-SIGN/L-SIGN ECD sample or 120 μL of Langerin/MGL ECD sample were injected onto the surfaces at 20 μL/min flow rate. Surface regeneration was performed by injecting 10 μL of 50 mM EDTA, demonstrating that the CLR/Spike interaction is strictly $Ca^{2+}$ dependent and confirming the specific recognition of glycans. The resulting sensorgrams were reference surface corrected (subtractions from Fc1 signal). The apparent affinity of compounds was determined by fitting the steady state affinity model to the plots of binding responses versus concentration.

## Cell lines

Baby hamster kidney cells (BHK-21, 12-14-17 MAW, Kerafast, Boston, MA) and African Green Monkey Cell Line (Vero E6) were cultured in Dulbecco´s modified Eagle medium (DMEM) supplemented with 10% heat-inactivated fetal bovine serum (FBS), 25 μg/mL gentamycin and 2 mM L-glutamine. Calu-3 cell line were kindly provided by Luis Enjuanes (CNB, CSIC, Madrid, Spain) and were cultured in DMEM supplemented with 20% heat-inactivated FBS, 25 μg/mL gentamycin and 2 mM L-glutamine. Jurkat, Jurkat DC-SIGN, Jurkat L-SIGN [9] and Jurkat Langerin were maintained in RPMI 1640 supplemented with 10% heat-inactivated FBS, 25 μg/mL gentamycin and 2 mM L-glutamine.

## Production of human monocyte-derived macrophages and dendritic cells

Blood samples were obtained from healthy human donors (Hospital 12 de Octubre, Madrid, Spain) under informed consent and IRB approval. Peripheral blood mononuclear cells (PBMCs) were isolated by Ficoll density gradient centrifugation (Ficoll Paque, GE17-5442-02, Sigma-Aldrich). To generate monocyte-derived dendritic cells (MDDCs) and monocyte-derived macrophages (M2-MDMs), CD14+ monocytes were purified using anti-human CD14 antibody-labeled magnetic beads and iron-based LS columns (Miltenyi Biotec) and used directly for further differentiation into dendritic cells and macrophages (Dominguez-Soto et al., J Immunol 2011). Differentiation to immature MDDCs was achieved by incubation at 37˚C with 5% CO2 for 7 days and subsequent activation with cytokines GM-CSF (1000 U/mL) and IL-4 (500 U/mL) (Miltenyi Biotec) every other day. For differentiation of M2-MDMs, cells were incubated at 37˚C with 5% CO2 for 7 days and activated with M-CSF (1000 U/mL) (Miltenyi Biotec) every second day.

## Production of SARS-CoV-2 pseudotyped recombinant Vesicular Stomatitis Virus (rVSV-luc)

rVSV-luc pseudotypes were generated following a published protocol (Whitt, J Virol Methods 2010). First, BHK-21 were transfected to express the S protein of SARS-CoV-2 (codon optimized, kindly provided by J. García-Arriaza, CNB-CSIC, Spain), Ebola virus Makona Glycoprotein (EBOV-GP) (KM233102.1) was synthesized and cloned into pcDNA3.1 by GeneArt AG technology (Life Technologies, Regensburg, Germany) or VSV-G following the manufacturer's instructions of Lipofectamine 3000 (Fisher Scientific). After 24 h, transfected cells were inoculated with a replication-deficient rVSV-luc pseudotype (MOI: 3–5) that contains firefly luciferase instead of the VSV-G open reading frame, rVSVΔG-luciferase (G*ΔG-luciferase) (Kerafast, Boston, MA). After 1 h incubation at 37˚C, cells were washed exhaustively with PBS and then DMEM supplemented with 5% heat-inactivated FBS, 25 μg/mL gentamycin and 2 mM L-glutamine were added. Pseudotyped particles were collected 20–24 h post-inoculation, clarified from cellular debris by centrifugation and stored at -80˚C [20,21,95]. Infectious titers were estimated as tissue culture infectious dose per mL by limiting dilution of rVSV-luc-

pseudotypes on Vero E6 cells. Luciferase activity was determined by luciferase assay (Steady-Glo Luciferase Assay System, Promega).

### Direct infection with pseudovirus

Cell lines: Jurkat, Jurkat DC-SIGN and Jurkat L-SIGN ($3 \times 10^5$ cells) or primary cells: MDDCs, M2-MDMs ($5 \times 10^4$ cells) were challenged with SARS-CoV-2, EBOV-GP or VSV-G pseudo-typed recombinant viruses (MOI: 0.5–2). After 24 h of incubation, cells were washed twice with PBS and lysed for luciferase assay.

### Trans-infection with pseudovirus

For trans-infection studies, Jurkat, Jurkat DC-SIGN, Jurkat L-SIGN, Jurkat Langerin ($3 \times 10^5$ cells) or MDDCs ($5 \times 10^4$ cells) were challenged with recombinant SARS-CoV-2, EBOV-GP or VSV-G pseudotyped viruses (MOI: 0.5–2) and incubated during 2 h at room temperature with rotation. Cells were then centrifuged at 1200 rpm for 5 minutes and washed twice with PBS supplemented with 0.5% bovine serum albumin (BSA) and 1 mM $CaCl_2$. Jurkat, Jurkat DC-SIGN, Jurkat L-SIGN and Jurkat Langerin or MDDCs were then resuspended in RPMI medium and co-cultivated with adherent Vero E6 cells ($1.5 \times 10^5$ cells/well) on a 24-well plate. After 24 h, the supernatant was removed and the monolayer of Vero E6 was washed with PBS three times and lysed for luciferase assay. The inhibitory effect of the glycomimetic compound PM26 on DC-SIGN-mediated trans-infection by Jurkat DC-SIGN to susceptible Vero E6 cells was tested in the same way as for authentic SARS-CoV-2 isolate (see below). PM26 was tested in triplicate and the IC50 value estimated using GraphPad Prism v8 with a 95% confidence interval and settings for normalize dose-response curves.

### The authentic SARS-CoV-2 culture

The authentic SARS-CoV-2 virus (clinical isolate 213196167, passage 2, Hospital Universitario 12 de Octubre, Madrid, Spain, GISAID reference EPI_ISL_1120962) was cultured in Vero E6 cells in 25 $cm^2$ cell culture flasks until cytopathic effect was observed. The supernatant containing the virus was then collected, centrifuged at 1200 rpm for 10 min to remove cell debris and stored at -80˚C for further experiments. The 50% tissue culture infectious dose ($TCID_{50}$) of the authentic SARS-CoV-2 was determined following previously described protocol [96]. Briefly, Vero E6 cells ($2 \times 10^4$ cells/well) seeded in a 96-well cell culture plate were incubated with 10-fold serial dilution of the authentic SARS-CoV-2 in DMEM supplemented with 5% FBS. Each dilution of the virus was tested by 6 replicates. After 48h of incubation at 37˚C, the supernatant was removed and Vero E6 were washed with PBS twice. The cells were then fixed with methanol containing 10% paraformaldehyde for 24h at 4˚C. The next day, the fixing buffer was removed and the cells were washed with 200 μl of PBS. Vero E6 were then permeabilized by adding 150 μl of PBS containing 0.1% Triton X-100 for 15min at room temperature. The cells were then washed with PBS three times and blocked by adding 200 μl of blocking solution (3% non-fat milk in PBS containing 0.1% Tween 20, PBST) for 1h at room temperature. After blocking, 100 μl of rabbit anti SARS-CoV Nucleocapsid antibody (#200-401-A50, Rockland Immunochemicals, Inc.) diluted 1:1000 in 1% non-fat milk in PBST was added to all wells and the plates were incubated for 1h at room temperature. The plates were then washed with PBS three times. Next, 100 μl of anti-rabbit IgG, HRP-linked antibody (#7074, Cell Signaling Technology) diluted 1:1000 in 1% non-fat milk in PBST was added to all wells and the plates were incubated again for 1h at room temperature. The plates were then washed with PBS three times and 100 μl of SIGMAFAST OPD (Sigma–Aldrich) was added. Following 10 min incubation time at room temperature, the reaction was stopped by adding 50 μl of 3M

hydrochloric acid. The optical density at 490 nm (OD490) was measured using a Thermo Scientific Multiskan FC plate reader (Thermo Fisher Scientific). The positive signal was selected when exceeded an OD490 of the mean of negative controls plus 3 standard deviations of the negative controls. The $TCID_{50}$ of the authentic SARS-CoV-2 clinical isolate was calculated according to the Reed-Muench method. All assays including the authentic SARS-CoV-2 isolate were performed in a BSL3 facility at Instituto de Investigación Hospital 12 de Octubre.

### Trans-infection with the SARS-CoV-2 clinical isolate

For trans-infection experiment, Calu-3 ($2 \times 10^4$ cells/well) were seeded in a 96-well cell culture plate. The following day, MDDCs ($2 \times 10^4$ cells) or Jurkat and Jurkat DC-SIGN ($2 \times 10^4$ cells) were challenged with the authentic SARS-CoV-2 at MOI 0.1–1. Anti-DC/L-SIGN antibody (MAB16211, R&D Systems) was used as a control of DC/L-SIGN-dependent infection. In case of using the anti-DC/L-SIGN antibody or compound PM26, the cells were first pre-incubated 20 min with the antibody or the compound before being challenged with authentic SARS-CoV-2. PM26 was tested at 3 different concentrations: 5 μM, 500 nM and 50 nM in case of MDDC and at 2 concentrations: 5 μM and 500 nM in case of Jurkat DC-SIGN. Cells were then incubated with the SARS-CoV-2 clinical isolate during 2 h at room temperature with rotation. Cells were then centrifuged at 1200 rpm for 5 minutes and washed with PBS supplemented with 0.5% BSA and 1mM $CaCl_2$ three times. Cells were then resuspended in RPMI medium and co-cultivate with adherent Calu-3 cells. After 48 h, the supernatant was removed and monolayer of Calu-3 was washed with PBS three times. The trans-infection in Calu-3 was then estimated by applying the protocol described above. Briefly, the cells were fixed, permeabilized, blocked with 3% non-fat milk, incubated with rabbit anti SARS-CoV Nucleocapsid antibody (1:1000), washed and incubated with anti-rabbit IgG-HRP antibody (1:1000). The plates were developed using SigmaFast OPD and the absorbance was measured at 490 nm using a Thermo Scientific Multiskan FC plate reader. The trans-infection was measured by 6–12 replicates and the data were analyzed by GraphPad Prism V8.

### Synthesis of Inhibitors

Polyman26 (PM26) is a known glycomimetic ligand of DC-SIGN and an antagonist of DC-SIGN mediated HIV trans-infection [49,50]. It was synthesized as previously described and tested in SPR studies as an inhibitor of DC-SIGN interaction to the spike protein of SARS-CoV-2, using both the oriented and non-oriented S surface described above. In both cases, a 20 μM solution of DC-SIGN in a running buffer composed of 25 mM Tris pH 8, 150 mM NaCl, 4 mM $CaCl_2$, 0.05% P20 surfactant was co-injected with variable concentrations of Polyman26, from 50 μM to 0.1 μM, in the same buffer. $IC_{50}$ values were determined from the plot of PM26 concentration vs % inhibition by fitting four-parameter logistic model as previously described [97].

### Supporting information

**S1 Fig. Chromatograms of Superose 6 elution of SARS-CoV-2 Spike (A), DC-SIGN ECD (B) and mix of DC-SIGN ECD and SARS-CoV-2 Spike (C).** In panel C, peak at 68 mL correspond to DC-SIGN ECD, which is in excess, and shouldered peak from 40 to 60 mL correspond to spike protein interacting with various numbers of DC-SIGN ECD. Fractions used for Electron Microscopy micrographs of DC-SIGN/SARS-CoV-2 Spike protein complexes (Fig 3C) are represented by the gray square (48 to 50 mL).
(TIF)

**S2 Fig. Sensorgrams of DC-SIGN/S protein interaction inhibition by a range of PM26 concentration.** In panel C, peak at 68 mL correspond to DC-SIGN ECD, which is in excess, and shouldered peak from 40 to 60 mL correspond to spike protein interacting with various numbers of DC-SIGN ECD. Fractions used for Electron Microscopy micrographs of DC-SIGN/SARS-CoV-2 Spike protein complexes (Fig 3C) are represented by the gray square (48 to 50 mL).
(TIF)

**S3 Fig. DC-SIGN expression in MDDCs by flow cytometry analysis.** The colored cell population represents MDDCs stained with antibody Anti-DC-SIGN_PE. In grey unstained MDDCs. DC-SIGN expression was analyzed by flow cytometry in a BD FACS Canto II Cell Analyzer with FlowJo V10.7.1 software.
(TIF)

## Acknowledgments

F.F. thanks warmly J. McLellan for sharing the spike expressing vectors and E. Fadda for making models of the spike glycoprotein available as well as for stimulating exchanges on the #glycotime on twitter.

## Author Contributions

**Conceptualization:** Rafael Delgado, Franck Fieschi.

**Funding acquisition:** Rafael Delgado, Franck Fieschi.

**Investigation:** Michel Thépaut, Joanna Luczkowiak, Corinne Vivès, Nuria Labiod, Fátima Lasala, Daphna Fenel.

**Resources:** Isabelle Bally, Yasmina Grimoire, Nicole Thielens, Guy Schoehn, Anna Bernardi.

**Supervision:** Anna Bernardi, Rafael Delgado, Franck Fieschi.

**Visualization:** Michel Thépaut, Joanna Luczkowiak, Corinne Vivès, Guy Schoehn.

**Writing – original draft:** Rafael Delgado, Franck Fieschi.

**Writing – review & editing:** Sara Sattin, Anna Bernardi, Rafael Delgado, Franck Fieschi.

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
