## [Decision Letter · Decision Letter 0]

21 Dec 2020

Dear Pr FIESCHI,

Thank you very much for submitting your manuscript "DC/L-SIGN recognition of spike glycoprotein promotes SARS-CoV-2 trans-infection and can be inhibited by a glycomimetic antagonist" for consideration at PLOS Pathogens. As with all papers reviewed by the journal, your manuscript was reviewed by members of the editorial board and by several independent reviewers. In light of the reviews (below this email), we would like to invite the resubmission of a significantly-revised version that takes into account the reviewers' comments.

You'll see that the reviewers were supportive of the work and had constructive comments to make. Two important critiques span the three reviewers: 1) the need to use authentic SARS-CoV-2 in the trans-infection models and 2) additional data on the transient internalization hypothesized to be part of the mechanism through which trans-infection takes place. You should consider and respond to the other comments of course, but those two are critical to address in a revised manuscript.

We cannot make any decision about publication until we have seen the revised manuscript and your response to the reviewers' comments. Your revised manuscript is also likely to be sent to reviewers for further evaluation.

Sincerely,

Andrew Pekosz, Ph.D.

Section Editor

PLOS Pathogens

Andrew Pekosz

Section Editor

PLOS Pathogens

Kasturi Haldar

Editor-in-Chief

PLOS Pathogens

orcid.org/0000-0001-5065-158X

Michael Malim

Editor-in-Chief

PLOS Pathogens

orcid.org/0000-0002-7699-2064

You'll see that the reviewers were supportive of the work and had constructive comments to make. Two important critiques span the three reviewers: 1) the need to use authentic SARS-CoV-2 in the trans-infection models and 2) additional data on the transient internalization hypothesized to be part of the mechanism through which trans-infection takes place. You should consider and respond to the other comments of course, but those two are critical to address in a revised manuscript.

Reviewer's Responses to Questions

**Part I - Summary**

Reviewer #1: This is a strong study which mixes multiple experimental approaches to suggest that DC-SIGN and L-SIGN are important attachment receptors which mediate SARS-CoV-2 trans infection. EM, surface plasmon resonance, bioinformatics, and cell culture models all complement each other nicely.

I would have liked some data if available showing the location of the virus prior to trans infection. As mentioned in the manuscript there is evidence that the virus is contained in a non-endolysosomal compartment in the dendritic cell prior to trans-infection. Did the authors consider EM experiments to demonstrate this?

Reviewer #2: Thépaut M et al., propose that DC/L-SIGN, two C-type lectin receptors (CLRs), present in immune cells such as macrophages and dendritic cells in the mucosa and respiratory tissue, serve as attachment points for SARS-CoV-2. These CLRs could recognize glycan sites on Spike. While these immune cells are not infected, they can direct the attached viruses to the surrounding cells that are then infected by a process called trans-infection. The authors showed by surface plasmon resonance (SPR) that when put together, purified DC-SIGN and L-SIGN can associate with Spike from SARS-CoV-2. Electron Microscopy experiments showed the complexes formed by DC-SIGN and Spike and by LC-SIGN and Spike, which are, most of the time, at a stoichiometric ratio of 1:1. Monocyte-derived dendritic cells (MDDCs), and M2 monocyte derived macrophages (M2-MDM), which are cells known to express DC/L-SIGN, were used for the trans-infection experiments, as well as modified Jurkat cells expressing or not DC/L-SIGN. These cells were incubated with VSV/SARS-CoV-2 pseudotyped viruses, that contains firefly luciferase, and then co-cultured with Vero E6 cells. Trans-infection in Vero E6 cells was monitored by the measure of luciferase activity. Authors showed that upon co-culture, pseudoviruses were able to infect Vero E6 cells and this was inhibited using an anti-DC/L-SIGN antibody, suggesting that cells expressing these CLRs can indeed bind the virus and carry it to other cells that can be infected. Moreover, interaction of DC-SIGN with Spike was also inhibited with Polyman26 (PM26), a multivalent glycomimetic mannoside and antagonist of DC-SIGN, which confirms that DC-SIGN recognizes glycans on Spike, as suggested by the authors. Finally, PM26 was also able to inhibit trans-infection on Vero E6 cells.

The subject of this research is of high interest because of the actual pandemic and the potential use of inhibitors of a potential trans-infection phenomenon occurring in the lungs. However, the trans-infection experiments should have been done with a more physiological cell model. Non-human Vero E6 cells are not advised to evaluate SARS-CoV-2 infection of the host. Respiratory cell types of human origin would be an appropriate model. Also, while experiments with pseudotyped viruses have allowed us to learn a lot about SARS-CoV-2 infection, isolated viruses would have also been a better model, in particular for experiments based on the interaction of the virion with the cell.

Reviewer #3: In this study, Thépaut and colleagues report a potential contribution of C-type lectin receptors in SARS-CoV-2 trans-infection of ACE2-expressing cells. For this, they expressed and purified a soluble SARS-CoV-2 Spike (S) stabilized ectodomain protein and performed SPR experiments using soluble multimeric C-type lectin receptors (DC-SIGN, L-SIGN, MGL, and langerin). They found that they all bind with the S ectodomain to different extent and that, although not functioning as entry/triggering receptors, the C-type lectins can mediate trans-infection of VSV pseudotypes from antigen presenting cells to susceptible Vero E6 cells. Although the study is interesting and reveal a potentially important mechanism of infection, the study falls short in characterizing the interaction of SARS-CoV-2 S with C-type lectin receptors and comparing to other viral glycoproteins, and the mechanism by which trans-infection occurs.

**Part II – Major Issues: Key Experiments Required for Acceptance**

Reviewer #1: No further experiments are absolutely needed. Please see the above. I think the data as shown demonstrate that like SARS, trans-infection, mediated by DCL-SIGN, is something that does occur, at least in vitro.

Reviewer #2: 1- First, all experiments are performed with pseudoviruses as a surrogate of virus infection. The authors are interested in DC/L-SIGN as potential attachment points for SARS-CoV-2 (not as an entry receptor). It is known that the virus envelope or other proteins present on the surface may impact the interaction of the virus with host cells. N-glycan sites are not restricted to Spike and could play a role in interaction with DC/L-SIGN proteins. Use of Spike mutant impairing glycosylation are required to strengthen the conclusion that DC/L-SIGN interacts with glycans on Spike. Additionally, it is difficult to know at which extend the results presented in this study are relevant to an infection with the whole virus. If possible, some of the experiments should be done with isolated infectious SARS-CoV-2 virus?

2- The second major comment is related to the use of Vero E6 cells for the trans-infection experiments. Vero E6 are not an appropriate model for this particular type of study. Particularly, the entire paper, the authors refer to dissemination of the infection in the respiratory tissue. If the authors want to claim demonstration of dissemination of the infection in respiratory tissue, key experiments should be done at a minimum using a respiratory cell line (Calu-3, A549 expressing ACE2, or other respiratory cell lines expressing ACE2) or event better primary cells.

If not possible, the conclusions and phrasing of the interpretation need to be revised. The scope of the current results is less significant than claimed by the authors.

3- The authors state that DC/L-SIGN “can also internalize viral particles into cells for storage in non-lysosomal compartments and subsequent transfer to susceptible cells in the process recognized as trans-infection”. The authors claim that it is the main process involved in the trans-infection observed in their study between immune cells and Vero E6 cells. However, the authors do not show evidence of viral particles internalization. Experiment showing that this is really happening are required to support this claim.

Reviewer #3: One major shortcoming of this study is the lack of validation with native virus. Indeed, the glycosylation of S onto native virions might be different than the recombinant secreted ectodomain and even from VSV pseudotypes. For instance, the formation of the membrane-associated viral factories and budding at the ERGIC followed by exocytosis might have an effect on glycosylation that could translate in differential binding to C-type lectin receptors.

In addition, the SPR data are difficult to reconcile, especially given the difference in binding for DC-SIGN with the non-oriented and oriented Spike ectodomain. The Kd values are relatively high and the lack of saturation for Langerin is problematic. Controls to with soluble ACE2 to validate the S ectodomain protein, as well as deglycosylated S to confirm that interaction with the C-type lectin receptors is specifically glycan dependent (may find that Langerin binding is non-specific, explaining the lack of saturation). In addition, comparison of SARS-CoV-2 and SARS-CoV ectodomain would enhance the study, or as an alternative, comparison to EBOV GP ectodomain could be done and correlated to the trans-infection data.

Another issue is that the outcome of C-type lectin receptor-S binding is unknown as well as the mechanism by which trans-infection occurs. For instance, once there is C-type lectin engagement, what happens to the virions? Are they internalized? Does this lead to some degradation/inactivation of a fraction of the viral population? Are the C-type lectin receptors activated? Is trans-infection more efficient than cell-free infection? Is this unique to SARS-CoV-2 when compared to SARS-CoV?

**Part III – Minor Issues: Editorial and Data Presentation Modifications**

Reviewer #1: I would perhaps discuss a bit more the possible location of the virus in the dendritic cell prior to trans infection. This is mentioned briefly but could help the reader understand the process a bit more. For HIV there is a debate about whether the virus simply sticks to the surface of the cell, or is actively pinocytosed into a non-endosomal compartment awaiting trans-infective release.

Reviewer #2: In the text, the authors state that they could not calculate kinetic association and dissociation rate constants (kon and koff). An inverted experiment, i.e. attach the lectin receptors to the surface and inject the spike protein, could help solve the problem

“Our work shows that DC/L-SIGN are important enhancers of infection mediated by the S protein of SARS-CoV-2 that greatly facilitate viral transmission to susceptible cells.” The term “enhancers” is not the appropriated term here. Authors are not showing a “basal” infection and then the enhancement of it by the presence of DC/L-SIGN. They show trans-infection, but not an enhancement of the infection.

Histograms with SEM of all figures should be represented as scatter plots with each point representing one individual replicate for better appreciation of variations.

The authors conclude “the specific topological presentation of their CRD as well as their oligomeric status is preserved for each of the CLR”. This conclusion is based on the use of recombinant constructs corresponding to the extracellular domains. Additional experiments are needed to verify if these properties are retained in the entire protein or reference that support this conclusion is needed. If not, this sentence should be rephrased.

Fig1: Transmission Electron Microscopy images in G. Is there some protein aggregation at this time point? If so, could the authors confirm that for these studies, proteins at day 22 were not used?

In a supplementary figure, the authors should show what is described in the text, “twenty out of the twenty-two SARS-CoV-2 S protein N-linked glycosylation sequons” conserved, when compared to SARS-CoV-1 S.

Fig2:

- Negative controls should be shown as supplementary information. Ex: Lectins alone, do they attach to the surface?

- The interaction of Spike and DC/L-SIGN is indeed verified here in vitro, which is the main point of the figure. If possible this result should be confirmed using pseudotyped viruses.

Fig3:

- In A and B, the models of the two proteins observed by TEM are represented. It would be interesting to also show a model of the complex that is observed by TEM in C.

- In the text the authors say they are expecting a stoichiometric Spike/CLR complex of 1:1, but for the TEM experiments they use a ratio of 1:3 (1 trimeric spike for 3 tetrameric DC-SIGN ECD). Is there a specific reason for this? If so, this should be commented in the discussion. This information might be useful for other people who would want to do similar experiments.

- SEC profiles should be shown as supplemental figures. It is valuable information for other researchers who might want to do experiments related to these results.

Fig4:

- The results presented in this figure should be repeated in a more relevant model: Calu-3, A549 expressing ACE2, or other respiratory cell lines expressing ACE2.

- In panel A, authors used MDDC and M2-MDC cells, while in panel B, the trans-infection is shown only in MDDC cells. Were experiments in M2-MDC done as well? If so, results should be added here. In particular, as in the discussion, authors conclude on both cell lines. If there are results on M2-MDC, they should appear in Fig. 4, along with MDDC trans-infection. If not, the conclusion (lines 533-536) needs to be restricted to MDDCs.

- Data showing DC/L-SIGN expression in these cells should be provided.

- More details should be provided in the legend to understand the figure.

Fig5:

- More details should be provided in the legend to understand the figure.

- Line 377: “Interestingly, no trans-infection was observed using Jurkat Langerin cells.” Why is it interesting? This was expected from the SRP results.

- Sentence in the discussion “On the other hand, DC/L-SIGN expression on Jurkat cells allows binding and capture of SARS-CoV-2 pseudovirions”. There is no experiment showing the binding and capture of pseudovirions in these cells.

Fig6:

- Another major comment: this is one of the most important figures of the paper, or at least it is important for the conclusion about using PM26 as an inhibitor to decrease the spread of the infection. However, the experiments related to the inhibition (panels C and D) seem to have been performed only once. These results must be replicated in at least 3 independent experiments.

- Authors use PM26 as an antagonist for DC-SIGN. Is there a similar mannoside for L-SIGN. If so, it would be relevant to perform similar experiment with it. In particular, as L-SIGN seems to show a better trans-infection rate than DC-SIGN (Fig 5C).

- It is written that the oriented and non oriented Spike set-ups were used for the SPR competition assays, but in panel B, there is only the oriented one. sensorgrams of the interaction with the non-oriented set-up should be added.

- Some details are missing to entirely understand the figure without extensively refering to the material and methods or results sections.

In the discussion, it is said that type II alveolar cells are known to express high levels of L-SIGN. If type II alveolar cells, believed to express ACE2, express also L-SIGN, would there be a trans-infection phenomenon here, or L-SIGN could help to attract and bind Spike, so then ACE2 could properly bind the latter? Maybe something to add to the discussion.

Lines 565-567: “DC/L-SIGN expression can be upregulated as well, since it has been demonstrated that while innate immune responses are potently activated by SARS CoV-2, it also counteracts the production of type I and type III interferon”. The sentence is confusing. Has it been shown that DC/L-SIGN is upregulated by SARS-CoV-2 infection or is it an assumption from the fact that IFN-I and III are downregulated during a SARS-CoV-2 infection?

Lines 584-587: “Additionally, upon binding to DC-SIGN PM26 was shown to induce a pro-inflammatory anti-viral response, which should be beneficial at the onset of the infection and may help to steer the immune response towards a profile correlated 586 with milder forms of the disease.” Either there is something wrong with the sentence or a deeper discussion of this idea needs to be provided. As mentioned by the authors, the cytokine storm is associated with SARS-CoV-2 pathogenesis. It is therefore difficult to understand why one would use a pro-inflammatory molecule in patients that would already have an initiated proinflammatory response?

Final paragraph of the discussion “Demonstration of involvement of APCs in SARS-CoV-2 early dissemination through CRLs DC/L-SIGN…”. This is clearly an overstatement. There is no demonstration of this in this study. The presented results may at most suggest a model. As discussed above further, more physiologically relevant experiments are required to draw such statements.

The last sentence of the abstract should be modified: the authors should use respiratory cell lines and not Vero E6 cells to conclude on a potential spreading of infection.

Minor comments related to the format:

There are some sentences/ideas where references are missing: Lines 67-75, line 107 (transmission rates), lines 110-117, lines 431-437, lines 446-448, lines 455-457, lines 463-465, lines 498-499, lines 515 (HIV).

Unformatted reference at line 573-574.

Line 562: replace though by through?

Line 581: replace Vero cells by Vero E6 cells.

Use C-type lectin receptors (CLR) abbreviation throughout the text after first definition.

Definition of DC or L-SIGN abbreviation was not introduced.

Sentence at lines 99-101 reads oddly. Do the authors mean that the cleavage at S2’ site is necessary to trigger conformational changes in S2? If so, I think it is difficult to understand that idea with the sentence as it is written.

Sentence at line 493: “DC-SIGN can also bind to some o complex N-glycosylation sites”. What does it mean some "o" complex?

Reviewer #3: In the introduction, the authors hypothesized that difference in binding of attachment factors (such as C-type lectin receptors), may play a role in differential transmission rate of SARS-CoV-2 when compared to SARS-CoV (lines 107-110). This led me to think that this study will provide sought-after comparative data of SARS-CoV and SARS-CoV-2, yet only SARS-CoV-2 is investigated here, and this study does test the hypothesis. As mentioned above, data with SARS CoV ectodomain would greatly enhance this study. The mention of the authors’ hypothesis puts the spotlight on the absence of data with SARS-CoV.

In addition, although I agree with the authors that the use of the full ectodomain is more likely to give biologically relevant data than the RBD alone (lines 462-463), it is important to note that an uncleaved and stabilized version was used and some caution should be taken when interpreting the data. Was a stabilized version of the spike used for the VSV pseudotypes? This is not mentioned in the materials and methods.

In line 137, virulence is inaccurate, at least for the D614G variant (better at replicating and infecting upper airway epithelial cells, potentially better transmitted, but no clear effect on virulence). Please provide references for an effect on virulence.

Please, provide p values for the infection data (Figs.4-6).

Some sections of the discussion are too speculative and are not based on the data presented (Lines 561-576).

Line 78, add “enveloped” in front of viruses. This statement only applies to them.

PLOS authors have the option to publish the peer review history of their article (what does this mean?). If published, this will include your full peer review and any attached files.

Reviewer #1: No

Reviewer #2: No

Reviewer #3: No
---

## [Decision Letter · Decision Letter 1]

20 Apr 2021

Dear Pr FIESCHI,

We are pleased to inform you that your manuscript 'DC/L-SIGN recognition of spike glycoprotein promotes SARS-CoV-2 trans-infection and can be inhibited by a glycomimetic antagonist' has been provisionally accepted for publication in PLOS Pathogens.

Best regards,

Andrew Pekosz, Ph.D.

Section Editor

PLOS Pathogens

Andrew Pekosz

Section Editor

PLOS Pathogens

Kasturi Haldar

Editor-in-Chief

PLOS Pathogens

orcid.org/0000-0001-5065-158X

Michael Malim

Editor-in-Chief

PLOS Pathogens

orcid.org/0000-0002-7699-2064

There are some minor changes that should be made during the prepublication process but overall, the reviewers found the manuscript significantly improved.

Reviewer Comments (if any, and for reference):

Reviewer's Responses to Questions

**Part I - Summary**

Reviewer #1: My prior comments have been addressed.

Reviewer #2: The authors replied appropriately to the previous requests. The conclusions have been strengthened by the use of SARS-CoV-2 in Calu-3 cells.

Reviewer #3: This is a revision by Thépaut and colleagues. The authors have now provided data with replicative SARS-CoV-2 and in Calu-3, which were critical and strengthen the study. I just have a few small comments:

I appreciated the clarification provided by the authors regarding the SPR data (insignificant difference between Kd for DC-SIGN and high Kd values) in the response to the reviewers’ comments. I also understand the reluctance of the authors to perform additional controls given time constraints and I realize that some of the controls regarding the glycan-dependent interaction asked by me and reviewer 2 were viewed as “really routine treatment in SPR”. While I, again, understand that the authors may find using spike mutants with impaired glycosylation or deglycosylated forms unnecessary, the authors should at least add in the text the regeneration conditions (EDTA treatment) which are missing in the manuscript and use this opportunity to clarify that the Ca2+ dependence in binding is characteristic glycan interaction with CLRs indicating that the binding is glycan-dependent. The readers of PLoS Pathogens are not necessarily experts in lectin binding and/or SPR.

My other suggestion of comparing with SARS-CoV Spike was also not well received. I know that binding studies have been performed for SARS-CoV S with lectins, but there is no side-by-side comparison. In fact, my suggestion stems from the authors’ own statement early in the manuscript: “We posit that the enhanced transmission rate of SARS-CoV-2 relative to SARS-CoV-1 might result from a more efficient viral adhesion through host-cell attachment factors, which may promote efficient infection of ACE2+ cells.” (lines 111-114). Again, I understand the reluctance of the authors to perform experiments with the SARS-CoV S, but the fact that the authors are not testing their hypothesis while mentioning that the glycosylations are conserved between SARS-CoV-1/2 Spike (lines 130-132) is confusing. I urge the authors to modify the text.

**Part II – Major Issues: Key Experiments Required for Acceptance**

Reviewer #1: My prior comments have been addressed.

Reviewer #2: No additional comments

Reviewer #3: (No Response)

**Part III – Minor Issues: Editorial and Data Presentation Modifications**

Reviewer #1: My prior comments have been addressed.

Reviewer #2: N/A

Reviewer #3: In line 507: change non-permissive to non-susceptible

PLOS authors have the option to publish the peer review history of their article (what does this mean?). If published, this will include your full peer review and any attached files.

Reviewer #1: No

Reviewer #2: No

Reviewer #3: No

---

## [Editor Report · Acceptance letter]

30 Apr 2021

Dear Pr FIESCHI,

We are delighted to inform you that your manuscript, "DC/L-SIGN recognition of spike glycoprotein promotes SARS-CoV-2 trans-infection and can be inhibited by a glycomimetic antagonist," has been formally accepted for publication in PLOS Pathogens.

Best regards,

Kasturi Haldar

Editor-in-Chief

PLOS Pathogens

orcid.org/0000-0001-5065-158X

Michael Malim

Editor-in-Chief

PLOS Pathogens

orcid.org/0000-0002-7699-2064